# On the Universality of Graph Neural Networks on Large Random Graphs

**Nicolas Keriven**
CNRS, GIPSA-lab, Grenoble, France
`nicolas.keriven@cnrs.fr`

**Alberto Bietti**
NYU Center for Data Science, New York, USA
`alberto.bietti@nyu.edu`

**Samuel Vaiter**
CNRS, LJAD, Nice, France
`samuel.vaiter@cnrs.fr`

## Abstract

We study the approximation power of Graph Neural Networks (GNNs) on latent position random graphs. In the large graph limit, GNNs are known to converge to certain "continuous" models known as c-GNNs, which directly enables a study of their approximation power on random graph models. In the absence of input node features however, just as GNNs are limited by the Weisfeiler-Lehman isomorphism test, c-GNNs will be severely limited on simple random graph models. For instance, they will fail to distinguish the communities of a well-separated Stochastic Block Model (SBM) with constant degree function. Thus, we consider recently proposed architectures that augment GNNs with unique node identifiers, referred to as Structural GNNs here (SGNNs). We study the convergence of SGNNs to their continuous counterpart (c-SGNNs) in the large random graph limit, under new conditions on the node identifiers. We then show that c-SGNNs are strictly more powerful than c-GNNs in the continuous limit, and prove their universality on several random graph models of interest, including most SBMs and a large class of random geometric graphs. Our results cover both permutation-invariant and permutation-equivariant architectures.

## 1 Introduction

Graph Neural Networks (GNNs) are deep architectures defined over graph data that have garnered a lot of attention in recent years. They represent the state-of-the-art in many graph Machine Learning (graph ML) problems, and have been successfully applied to e.g. node clustering [7], semi-supervised learning [24], quantum chemistry [17], and so on. See [5, 44, 18, 21] for reviews.

As the universality of Multi-Layers Perceptrons (MLP) is one of the foundational theorems in deep learning – that is, any continuous function can be approximated arbitrarily well by an MLP – in the last few years the approximation power of GNNs has been a topic of great interest. In the absence of special node features, i.e. when one has only access to the graph structure, the crux of the problem has been proven to be the capacity of GNNs to solve the *graph isomorphism problem*, that is, deciding when two graphs are permutations of each other or not (a difficult problem for which no polynomial algorithm is known [4]). Indeed, this property is directly linked to the approximation power of GNNs [8, 3]. In this light, the landmark paper [45] proves that classical GNNs are at best as powerful as the famous Weisfeiler-Lehman (WL) test [43] for graph isomorphism. Since then, several works [45, 31, 8] have derived new architectures, for instance involving high-order tensors [31], with improved discriminative power equivalent to "higher-order" variants of the WL test. In another line of works, several recent papers have advocated the use of *unique node identifiers* [28, 27], with

35th Conference on Neural Information Processing Systems (NeurIPS 2021).

strategies to preserve the permutation-equivariance/invariance of GNNs coined Structural Message Passing (SMP) in [41] (see Sec. 2.3). We call these models Structural GNN (SGNN) here. SGNNs have been proved to be strictly more powerful than the WL test in [41], and even *universal* on graphs with bounded degrees, however for powerful layers that cannot be implemented in practice.

When the size of the graphs grows, the notion of graph isomorphism becomes somewhat moot: large graphs might share properties (number of well-connected communities, and so on) but are never isomorphic to each other. GNNs have nevertheless proven successful in identifying their large-scale structures, e.g. for node clustering [7]. Several papers have therefore used tools from random graph theory and graphons to study the behavior of GNNs *in the large-graph limit*. In [22, 38], GNNs are shown to converge to limiting "continuous" architectures (coined c-GNNs in [22]). A few works have studied the discriminative power of c-GNNs on graphons [30], however, analogously to how the WL test will fail on regular graphs, c-GNNs are severely limited on graph models with almost-constant degree function (Fig. 1), and the question is still largely open.

**Contribution and outline.** In this paper, we study the convergence of SGNNs on large random graphs towards "c-SGNNs", and analyze the approximation power of c-GNNs and c-SGNNs. After some preliminary results in Sec. 2, we study the convergence of SGNNs in Sec 3. In Sec. 4 and 5, we show that c-SGNNs are strictly more powerful than c-GNNs in both permutation-invariant and equivariant case. We then prove the universality of c-SGNNs on several popular models of random graphs, including Stochastic Block Models (SBMs) and radial kernels. For instance, we show in Sec. 5.3 that c-SGNNs can perfectly identify the communities of most SBMs, including some for which any c-GNN provably fails (Fig. 1). The code for the numerical illustrations is available at https://github.com/nkeriven/random-graph-gnn.

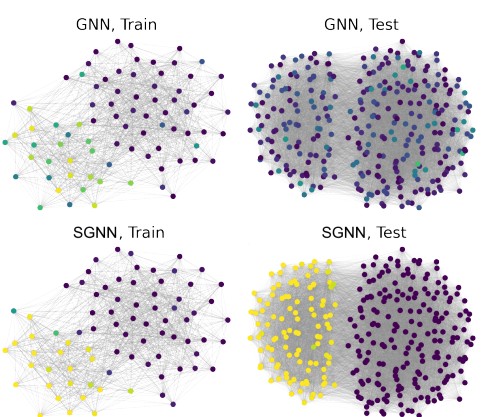

Figure 1: Illustration of Prop. 7 (Sec. 5.3). On an SBM with constant degree function, a GNN (top) might overfit the training set, but converges to a constant c-GNN. On the contrary, there exists a c-SGNN (bottom) that perfectly separates the communities. Details can be found in App. E.

**Related work.** The approximation power of GNNs (on finite graphs) has been a topic of great interest in recent years, and we do not attempt to make an exhaustive list of all results here. The landmark paper [45] showed that permutation-invariant GNNs were at best as powerful as the WL test, and later models [31, 8, 16] were constructed to be as powerful as higher-order WL tests, using for instance higher-order tensors [31]. The link between the graph isomorphism problem and function approximation was made in [8, 23] and extended in [3]. Some architectures were proven to be universal [32, 23] but involve tensors of unbounded order. When graphs are equipped with unique node identifiers, the approximation power of GNNs can be significantly improved if relaxing permutation-invariance/equivariance [28, 27, 10], or assuming that it holds only in expectation with respect to additional sources of randomness [39, 1]. On the contrary, as we will see in Sec. 2.3, the SGNN architecture proposed in [41] is inspired by the same ideas but still satisfies exact permutation-invariance/equivariance. As mentioned above, SGNNs are more powerful than the WL test [41], and even universal on graphs with bounded degrees when allowing powerful layers. To our knowledge, the approximation power of SGNNs as they are implemented in practice is still open. We treat of their continuous limit in this paper.

Fewer results can be found in the large-graph limit. Beyond the graph-isomorphism paradigm, authors have studied the capacity of GNNs to count graph substructures [6, 13] or identify various graph properties [15, 46]. Several works have studied the large-graphs limit of GNNs [38, 30, 26, 22], assuming random graph or graphon models. The degree function has been proven to be a crucial element for the discriminative power of c-GNNs, and they will not be able to distinguish graphons with close degree functions [30]. Here we show how c-SGNNs allow to overcome these limitations, and provide the first universality theorems for (S)GNNs in the continuous limit. Finally, we note that universal architectures exist on *measures* [11, 49], which can be seen as limits of *point clouds*, but, similar to the discrete case [48, 32, 23], the graph case may be significantly harder to study.

## 2 Preliminaries

In this section, we group notations on random graphs and GNNs, present the convergence result of [22] slightly adapted to our context, and introduce the SGNNs of [41]. An undirected graph $G$ with $n$ nodes is represented by a symmetric adjacency matrix $A \in \mathbb{R}^{n \times n}$. It is isomorphic to all graphs with adjacency matrices $\sigma A \sigma^\top$, for any permutation matrix $\sigma \in \{0, 1\}^{n \times n}$. Matrix rows (resp. columns) are denoted by $A_{i,:}$ (resp. $A_{:,i}$). The norm $\|\cdot\|$ is the operator norm for matrices and Euclidean norm for vectors. For a compact metric space $\mathcal{X}$, we denote by $\mathcal{C}(\mathcal{X}, \mathbb{R})$ the space of bounded continuous functions $\mathcal{X} \to \mathbb{R}$, equipped with $\|f\|_\infty = \sup_x |f(x)|$. For a multivariate function $f \in \mathcal{C}(\mathcal{X}, \mathbb{R}^d)$, $f_i$ denotes its $i$th coordinate, and $\|f\|_\infty = (\sum_i \|f_i\|_\infty^2)^{1/2}$.

### 2.1 Random graph model

We consider "latent position" random graphs, which include several popular models such as SBMs or graphons [29]. The nodes are associated with unobserved latent random variables $x_i$ drawn *i.i.d.* For the edges, we will examine two cases: an "ideal" one with deterministic weighted edges, and a more "realistic" one where the edges are randomly drawn independently. In the literature, the latter is often considered as an idealized framework to study the behavior of various algorithms [42, 37].

Let $(\mathcal{X}, m_\mathcal{X})$ be a compact metric space that is not a singleton, and assume that the $\varepsilon$-covering numbers[1] of $\mathcal{X}$ scale as $\mathcal{O}\left(\varepsilon^{-d_\mathcal{X}}\right)$ for some dimension $d_\mathcal{X}$. We denote by $\mathcal{W}$ the set of symmetric bivariate functions in $\mathcal{C}(\mathcal{X} \times \mathcal{X}, [0, 1])$ that are $L_W$-Lipschitz in each variable, and by $\mathcal{P}$ the set of probability distributions over $\mathcal{X}$ equipped with the total variation norm $\|\cdot\|_{\mathrm{TV}}$. A graph with $n$ nodes is generated according to a random graph model $(W, P) \in \mathcal{W} \times \mathcal{P}$ as follows:

$$x_1, \ldots, x_n \overset{i.i.d}{\sim} P, \qquad a_{ij} = \begin{cases} W(x_i, x_j) & \text{deterministic edges case} \\ \alpha_n^{-1}\mathrm{Bernoulli}(\alpha_n W(x_i, x_j)) & \text{random edges case} \end{cases}$$

where $\alpha_n$ is the *sparsity level* of the graph in the random edges case, which we assume to be known for simplicity. When $\alpha_n \sim 1$, the graph is said to be *dense*, when $\alpha_n \sim \frac{1}{n}$ the graph is *sparse*, and when $\alpha \sim \frac{\log n}{n}$ the graph is *relatively sparse*. Note that we have normalized the Bernoulli edges by $1/\alpha_n$ such that $\mathbb{E}a_{ij} = W(x_i, x_j)$ conditionally on $x_i, x_j$. When $\mathcal{X}$ is finite, the model is called an SBM (see Sec. 4.3).

Like finite graphs, random graph models can be isomorphic to one another [22, 29]. In this paper, similar to [22] we consider that, for any bijection $\varphi : \mathcal{X} \to \mathcal{X}$, the model $(W, P)$ is isomorphic to $(W_\varphi, \varphi_\sharp^{-1}P)$, where $W_\varphi(x, y) = W(\varphi(x), \varphi(y))$ and $f_\sharp P$ is the pushforward of $P$ (the distribution of $f(X)$). Indeed, it is easy to see that both produce exactly the same distribution over graphs.

### 2.2 Graph Neural Networks

Following [22, 12], in this paper we consider the so-called "spectral" version of GNNs, which include several message-passing models for certain aggregation functions. We consider polynomial filters $h$ defined as $h(A) = \sum_k \beta_k A^k$ for a matrix or operator $A$. In practice, the order of the filters is always finite, but our results are valid for infinite-order filters (assuming that the sum always converges for simplicity). We consider any activation function $\rho : \mathbb{R} \to \mathbb{R}$ which satisfies $\rho(0) = 0$ and $|\rho(x) - \rho(y)| \leqslant |x - y|$ for which the universality theorem of MLPs applies [35], e.g. ReLU.

Spectral GNNs are defined by successive filtering of a graph signal. Given an input signal $Z^{(0)} \in \mathbb{R}^{n \times d_0}$, at each layer $\ell = 0, \ldots, M - 1$:

$$Z^{(\ell+1)}_{:,j} = \rho\left(\sum_{i=1}^{d_\ell} h^{(\ell)}_{ij}\left(\tfrac{1}{n}A\right) Z^{(\ell)}_{:,i} + b^{(\ell)}_j 1_n\right) \in \mathbb{R}^n \qquad j = 1, \ldots, d_{\ell+1}, \tag{1}$$

where $h^{(\ell)}_{ij}(A) = \sum_k \beta^{(\ell)}_{ijk}A^k$ are trainable graph filters, $b^{(\ell)}_j \in \mathbb{R}$ are trainable additive biases and $\rho$ is applied pointwise. We note the normalization $A/n$ that will be necessary for convergence. GNNs exist in two main versions: so-called "permutation-invariant" GNNs $\bar\Phi$ output a single vector for the entire graph, while "permutation-equivariant" GNNs $\Phi$ output a graph signal:

$$\Phi_A(Z^{(0)}) = g\left(Z^{(M)}\right) \in \mathbb{R}^{n \times d_{out}} \qquad \bar\Phi_A(Z^{(0)}) = g\left(\tfrac{1}{n}\sum_{i=1}^n Z^{(M)}_{i,:}\right) \in \mathbb{R}^{d_{out}}, \tag{2}$$

---

[1]the number of balls of radius $\varepsilon$ required to cover $\mathcal{X}$

where $g : \mathbb{R}^{d_M} \to \mathbb{R}^{d_{out}}$ is an MLP applied row-wise in the equivariant case. By construction, such GNNs are indeed equivariant or invariant to node permutation: for all permutation matrices $\sigma$, we have $\Phi_{\sigma A \sigma^\top}(\sigma Z^{(0)}) = \sigma \Phi_A(Z^{(0)})$ and $\bar{\Phi}_{\sigma A \sigma^\top}(\sigma Z^{(0)}) = \bar{\Phi}_A(Z^{(0)})$.

**Continuous GNNs.** In [22], the authors show that, in the large random graphs limit, GNNs converge to the following "continuous" models (coined c-GNNs), which propagate **functions over the latent space** $f^{(\ell)} \in \mathcal{C}(\mathcal{X}, \mathbb{R}^{d_\ell})$ instead of graph signals. Given an input function $f^{(0)} \in \mathcal{C}(\mathcal{X}, \mathbb{R}^{d_0})$:

$$f_j^{(\ell+1)} = \rho \left( \sum_{i=1}^{d_\ell} h_{ij}^{(\ell)}(T_{W,P}) f_i^{(\ell)} + b_j^{(\ell)} \right),$$

where $T_{W,P}$ is the operator $T_{W,P} f = \int W(\cdot, x) f(x) dP(x)$. Then similar to (2) the permutation-equivariant/invariant versions of c-GNNs are defined as:

$$\Phi_{W,P}(f^{(0)}) = g \circ f^{(M)}, \qquad \bar{\Phi}_{W,P}(f^{(0)}) = g \left( \int f^{(M)}(x) dP(x) \right).$$

Remark that $\Phi_{W,P}(f^{(0)})$ is itself a *function* in $\mathcal{C}(\mathcal{X}, \mathbb{R}^{d_{out}})$ while $\bar{\Phi}_{W,P}(f^{(0)}) \in \mathbb{R}^{d_{out}}$ is a vector. Moreover, by virtue of the polynomial filters, the four architectures $\Phi_A, \bar{\Phi}_A, \Phi_{W,P}, \bar{\Phi}_{W,P}$ have the exact same set of parameters. Like in the discrete case, one can prove that c-GNN are equivariant or invariant to isomorphisms of random graph models [22]: for all bijection $\varphi : \mathcal{X} \to \mathcal{X}$, we have $\Phi_{W_\varphi, \varphi_\sharp^{-1} P}(f \circ \varphi) = \Phi_{W,P}(f) \circ \varphi$ and $\bar{\Phi}_{W_\varphi, \varphi_\sharp^{-1} P}(f \circ \varphi) = \bar{\Phi}_{W,P}(f)$.

Let us now turn to convergence of GNNs to c-GNNs. The following result is adapted[2] from [22]. While the outputs of permutation-invariant GNNs and c-GNNs are vectors in $\mathbb{R}^{d_{out}}$ that can be directly compared, the output graph signal of a permutation-equivariant GNN is compared with a *sampling* of the output function of the corresponding c-GNN: for a graph signal $Z \in \mathbb{R}^{n \times d}$, a function $f : \mathcal{X} \to \mathbb{R}^d$ and $X = \{x_i\}_{i=1}^n$, we define the (square root of the) mean-square error as $\mathrm{MSE}_X(Z, f) = (\frac{1}{n} \sum_{i=1}^n \|Z_i - f(x_i)\|_2^2)^{1/2}$. The proof of Theorem 1 with all multiplicative constants is given in App. A.1. Recall that $d_{\mathcal{X}}$ is the "dimension" of $\mathcal{X}$.

**Theorem 1.** *Assume $G$ is drawn from $(W, P)$ and has latent variables $X$. Fix $\rho, \nu > 0$.*

- *In the **deterministic edges case:** with probability $1 - \rho$, for all $Z^{(0)}$:*

$$\mathrm{MSE}_X(\Phi_A(Z^{(0)}), \Phi_{W,P}(f^{(0)})) \leqslant C \cdot \mathrm{MSE}_X(Z^{(0)}, f^{(0)}) + R_1(n) \qquad (3)$$

  *for some constant $C$ and $R_1(n) = \mathcal{O}\left( \sqrt{(d_{\mathcal{X}} + \log(1/\rho))/n} \right)$.*
- *In the **random edges case:** assume that the sparsity level is $\alpha_n \gtrsim n^{-1} \log n$. There is a constant $C_\nu$ such that, with probability $1 - \rho - n^{-\nu}$, for all $Z^{(0)}$:*

$$\mathrm{MSE}_X(\Phi_A(Z^{(0)}), \Phi_{W,P}(f^{(0)})) \leqslant C \cdot \mathrm{MSE}_X(Z^{(0)}, f^{(0)}) + R_1(n) + R_2(n) \qquad (4)$$

  *where $R_2(n) = \mathcal{O}\left( C_\nu / \sqrt{\alpha_n n} \right)$.*
- *In the **permutation-invariant case:** The exact same results hold for $\left\| \bar{\Phi}_A(Z^{(0)}) - \bar{\Phi}_{W,P}(f^{(0)}) \right\|$ instead of the MSE on the left-hand-side, with an added error term $R_3(n) = \mathcal{O}\left( \sqrt{\log(1/\rho)/n} \right)$.*

By the theorem above, a GNN converges to its continuous counterpart if $Z^{(0)}$ is (close to) a sampling of a function $f^{(0)}$ at the latent variables. This is directly assumed in [22]. In the present paper however, we do not suppose that input node features are available. While several strategies have been proposed in the literature, a popular baseline is to simply take constant input $Z^{(0)} = 1_n$ (which, by a multiplication by $A$ on the first layer, is also equivalent to inputing the degrees as in [7] for instance). In this case, there is convergence to a c-GNN with $f^{(0)} = 1$. For simplicity in the rest of the paper we drop the notation "(1)" and write $\Phi_A = \Phi_A(1_n)$, $\Phi_{W,P} = \Phi_{W,P}(1)$, and so on. It is known that GNNs are limited on regular graphs: for permutation-invariant GNNs, the WL test cannot distinguish regular graphs of the same order, and for permutation-equivariant GNNs, $\Phi_A(1_n)$ is *constant over the nodes*. Similarly for c-GNN, the degree function $\int W(\cdot, x) dP(x)$ is key in the discriminative power of c-GNNs [30], and if it is constant, then $\Phi_{W,P}(1)$ is a constant function (see Fig. 1).

---

[2]The authors in [22] proved this for the normalized Laplacian, which allows bypassing the knowledge of $\alpha_n$. Here we use the adjacency matrix, for our later results on approximation power.

## 2.3 SGNN: GNN with unique node identifiers

To remedy the absence of input node features, in [41] the authors propose an architecture with *unique node identifiers* while still respecting permutation invariance/equivariance. More precisely, they first choose an *arbitrary* ordering of the nodes $q = 1, \dots, n$, apply a GNN to each *one-hot vector* $e_q = [0, \dots, 1, \dots, 0]$, and restore equivariance to permutation by a final pooling. For later purpose of convergence, we generalize this strategy to any collection $E_q(A) \in \mathbb{R}^{n \times d_0}$ that satisfies:

$$E_q(\sigma A \sigma^\top) = \sigma E_{\sigma^{-1}(q)}(A). \tag{5}$$

where by an abuse of notation $\sigma(q)$ designates the permutation function applied to index $q$. For instance, it can be any filtering of one-hot vector $E_q(A) = h(A)e_q$. After the GNN $\Phi_A$, a pooling is applied to restore equivariance, then a second GNN $\Phi'_A$ that is either invariant or equivariant:

$$\Psi_A = \Phi'_A \left( \tfrac{1}{n} \sum_q \Phi_A \left( E_q(A) \right) \right) \in \mathbb{R}^{n \times d'_{out}}, \qquad \bar{\Psi}_A = \bar{\Phi}'_A \left( \tfrac{1}{n} \sum_q \Phi_A \left( E_q(A) \right) \right) \in \mathbb{R}^{d'_{out}} \tag{6}$$

In [41], these architectures, called SMPs, are interpreted as doing message-passing over matrices, and can use more general pooling and aggregation functions. In a sense, what we call SGNN are "spectral" versions of SMP, but are essentially the same idea. It is not difficult to see that SGNNs satisfy: $\Psi_{\sigma A \sigma^\top} = \sigma \Psi_A$ and $\bar{\Psi}_{\sigma A \sigma^\top} = \bar{\Psi}_A$.

## 3 Convergence of SGNNs on large random graphs

In this section, we extend Theorem 1 to SGNNs. To define continuous SGNNs, we consider a *bivariate* input function $\eta_{W,P} : \mathcal{X} \times \mathcal{X} \to \mathbb{R}^{d_0}$ such that: the mapping $(W, P) \mapsto \eta_{W,P}$ is continuous[3], for any $(W, P)$, $\eta_{W,P}$ is $C_\eta$-bounded and $L_\eta$-Lipschitz in each variable, and similar to (5) is respects the following:

$$\eta_{W_\varphi, \varphi_\sharp^{-1} P}(x, y) = \eta_{W,P}(\varphi(x), \varphi(y)), \tag{7}$$

for all bijections $\varphi : \mathcal{X} \to \mathcal{X}$. For instance, $\eta_{W,P} = W$ or any filter $\eta_{W,P}(x, y) = [h(T_{W,P})W(\cdot, y)](x)$ satisfy these conditions. A c-SGNN is then defined as:

$$\Psi_{W,P} = \Phi'_{W,P} \left( \int \Phi_{W,P}(\eta_{W,P}(\cdot, x)) dP(x) \right), \tag{8}$$

for the equivariant case, or similarly $\bar{\Psi}_{W,P} = \bar{\Phi}'_{W,P} \left( \int \Phi_{W,P}(\eta_{W,P}(\cdot, x)) dP(x) \right)$ for the invariant case. Again, using the properties of c-GNNs, it is easy to see that c-SGNNs respect random graphs isomorphism: $\Psi_{W_\varphi, \varphi_\sharp^{-1} P} = \Psi_{W,P} \circ \varphi$ and $\bar{\Psi}_{W_\varphi, \varphi_\sharp^{-1} P} = \bar{\Psi}_{W,P}$. We are now ready to extend Theorem 1. The proof and multiplicative constants are in App. A.2.

**Theorem 2.** *Assume $G$ is drawn from $(W, P)$ with latent variables $X$. Fix $\rho, \nu > 0$.*

- *In the deterministic edges case: with probability $1 - \rho$:*

$$\text{MSE}_X(\Psi_A, \Psi_{W,P}) \leqslant C' \cdot \sup_q \text{MSE}_X(E_q(A), \eta(\cdot, x_q)) + R'_1(n) \tag{9}$$

  *for some constant $C'$ and $R'_1(n) = \mathcal{O}\left( \sqrt{(d_\mathcal{X} + \log(1/\rho))/n} \right)$.*

- *In the random edges case: assume that the sparsity level is $\alpha_n \gtrsim n^{-1} \log n$. There is a constant $C_\nu$ such that, with probability $1 - \rho - n^{-\nu}$:*

$$\text{MSE}_X(\Psi_A, \Psi_{W,P}) \leqslant C' \cdot \sup_q \text{MSE}_X(E_q(A), \eta(\cdot, x_q)) + R'_1(n) + R'_2(n) \tag{10}$$

  *where $R'_2(n) = \mathcal{O}\left( C_\nu / \sqrt{\alpha_n n} \right)$.*

- *In the permutation-invariant case: The exact same results hold for $\left\| \bar{\Psi}_A - \bar{\Psi}_{W,P} \right\|$ instead of the MSE, with an added error term $R'_3(n) = \mathcal{O}\left( \sqrt{\log(1/\rho)/n} \right)$.*

Hence, we obtain convergence when the input signal $E_q(A)$ is close to being a sampling of a function $\eta_{W,P}$ at $x_q$. The choice of input signal/function is therefore crucial. Let us examine a few strategies.

---

[3] for the norm $\|\cdot\|_\infty + \|\cdot\|_{\text{TV}}$ on $\mathcal{W} \times \mathcal{P}$

**One-hot vectors.** If one chooses one-hot vectors $E_q = e_q$ as in [41], then we can see that the SGNN converges to the continuous architecture with input $\eta_{W,P} = 0$, since $\mathrm{MSE}_X(e_q, 0) \to 0$. Since $\Psi_{W,P}$ is nothing more than a traditional c-GNN $\Phi_{W,P}$ with constant input in that case, this is not a satisfying choice in terms of approximation power.

**One-hop filtering.** If we choose to "filter" $e_q$ once and take $E_q(A) = Ae_q$, the natural continuous equivalent is $\eta_{W,P} = W$. Such a strategy only works for deterministic edges: indeed,

$$\mathrm{MSE}_X(Ae_q, W(\cdot, x_q)) = \left(n^{-1} \sum_{i=1}^n (a_{iq} - W(x_i, x_q))^2\right)^{1/2}$$

$$\begin{cases} = 0 & \text{with deterministic edges} \\ \approx \sqrt{n^{-1} \sum_i Var(a_{iq})} \sim \alpha^{-1} & \text{with random edges, w.h.p.} \end{cases}$$

where the last line comes from a simple application of Hoeffding's inequality. Hence, in the case of random edges, the MSE does not vanish and typically diverges for non-dense graphs (Fig. 2).

**Two-hop filtering.** We can therefore choose to filter *twice* and consider $\eta_{W,P}(x, y) = T_{W,P}[W(\cdot, y)](x)$, that is, $E_q(A) = A^2 e_q / n$. We have the following result.

**Proposition 1.** *In the random edges case, with probability $1-\rho$, we have for all $q$:*

$$\mathrm{MSE}_X \left(\frac{A^2 e_q}{n}, T_{W,P}[W(\cdot, x_q)](\cdot)\right) \lesssim \mathcal{O}\left(\frac{\sqrt{\log(n/\rho)}}{\alpha_n \sqrt{n}}\right).$$

*In the deterministic edges case, the rate is $\mathcal{O}(1/\sqrt{n})$.*

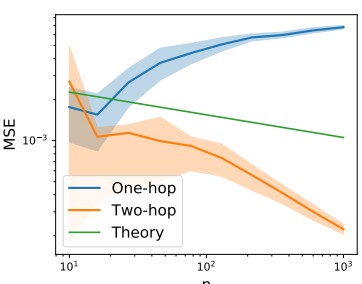

Figure 2: Difference between the outputs of some $\bar{\Psi}_A$ in the random edges case with $\alpha_n \sim n^{-1/3}$ and in the deterministic edges case, which converges to $\bar{\Psi}_{W,P}$. Convergence is observed for two-hop filtering only. The theoretical rate of Prop. 1 is slightly pessimistic. Details can be found in App. E.

Convergence is guaranteed when $\sqrt{\log n}/(\alpha_n \sqrt{n}) \to 0$, which is stronger than relative sparsity. The difference between one-hop and two-hop filtering is illustrated in Fig. 2. Other strategies may also be examined, and we leave this for future work. For instance, based on Theorem 5 in App. D, a strategy based on the eigenvectors of $A$ could lead to convergence even in the relatively sparse case. In the rest of the paper, we explicitly write when our results are valid for one- or two-hop filtering.

## 4 Approximation power: permutation-invariant case

In this section, we study the approximation power of continuous architectures in the permutation-invariant case. We seek to characterize the functions $\mathcal{W} \times \mathcal{P} \to \mathbb{R}$ that can be well-approximated by a c-GNN or c-SGNN. We derive a generic criterion for universality arising from the Stone-Weierstrass theorem, before proving that c-SGNNs are indeed strictly more powerful than c-GNNs. Finally, we give several examples of models for which c-SGNNs are universal.

### 4.1 Generic result with Stone-Weierstrass theorem

A classical tool to prove universality of neural nets in the literature is the Stone-Weierstrass theorem [20, 23], which states that an algebra of functions that separates points on a compact space is dense in the set of continuous functions. Although NNs are typically not an algebra since they are not closed by multiplication, one of the original proofs of universality for MLPs solves this by a clever trick [20] which we recall in Lemma 1 in App. D. A direct application results in the following proposition.

**Proposition 2.** *Let $\mathcal{M} \subset \mathcal{W} \times \mathcal{P}$ be a **compact** subset of $\mathcal{W} \times \mathcal{P}$. c-SGNNs (resp. c-GNNs) are dense in $\mathcal{C}(\mathcal{M}, \mathbb{R})$ if and only if: for all $(W, P) \neq (W', P') \in \mathcal{M}$, there is a c-SGNN (resp. a c-GNN) such that $\bar{\Psi}_{W,P} \neq \bar{\Psi}_{W',P'}$ (resp. $\bar{\Phi}_{W,P} \neq \bar{\Phi}_{W',P'}$).*

Note that, by construction of the permutation-invariant c-(S)GNNs, universality is only possible when $\mathcal{M}$ does *not* contain two isomorphic versions of the same random graph model. Equivalently, $\mathcal{M}$ may be a larger set *quotiented* by random graph isomorphism.

### 4.2 c-SGNNs are more powerful than c-GNNs

SGNNs were proven to be strictly more powerful than the WL test, and therefore than GNNs, in [41]. In the theorem below, we check that this strict inclusion holds for their continuous limits.

**Theorem 3.** *The set of functions of the form* $(W, P) \to \bar{\Phi}_{W,P}$ *is **strictly** included in the set of functions* $(W, P) \to \bar{\Psi}_{W,P}$*, for both one- and two-hop input filtering.*

This theorem is proven by constructing two models that are distinguished by a c-SGNN but not by any c-GNN. The principle is similar to the proof in the discrete case: in [41] the authors construct two $k$-regular graphs (with constant degree $k$), which by construction cannot be distinguished by the Weisfeiler-Lehman test, and prove that there is however a SGNN that distinguishes them. In the proof of 3, we construct two SBMs with the same constant degree function, such that any c-GNN returns the same result on them, and design a c-SGNN that distinguishes them. We note that there might be subsets $\mathcal{M} \subset \mathcal{W} \times \mathcal{P}$ that do not contain such models, and therefore on which c-GNNs and c-SGNNs have the same approximation power.

### 4.3 Examples

While a generic universality theorem on random graphs seems to be out-of-reach for the moment, we examine several interesting examples, and pave the way for future extensions. We focus on c-SGNNs, but, sometimes, may not conclude on the power of c-GNNs: we could not prove that they are universal, but were not able to find a counter-example either. For simplicity, and since our purpose here is mainly illustrative, we mostly focus on one-hop input filtering.

**Stochastic Block Models.** SBMs [19] are classical models to emulate graphs with communities. In our settings, SBMs with $K$ communities can be obtained with a finite latent space $|\mathcal{X}| = K$, typically $\mathcal{X} = \{1, \ldots, K\}$. The kernel $W(k, k') = W_{kk'}$ can be represented as a matrix $W \in S_K$, where $S_K$ is the set of symmetric matrices in $[0, 1]^{K \times K}$, and the distribution $P(k) = P_k$ as a vector $P \in \Delta^{K-1}$, where $\Delta^{K-1} = \{P \in [0, 1]^K, \sum_k P_k = 1\}$ is the $(K{-}1)$-dimensional simplex.

In the following proposition, we fix $P$ and examine universality with respect to $W$. In this case, continuous GNNs are actually quite similar to discrete ones on matrices $W$, except that the probability vector $P$ also intervenes in the computation. While GNNs on finite graphs can only be universal when using high-order tensors [32, 23] due to invariance to graph-isomorphism, here $P$ can help to disambiguate this constraint. We will say that $P \in \Delta^{K-1}$ is **incoherent** if: for signs $s \in \{-1, 0, 1\}^k$, having $\sum_{k=1}^K s_k P_k = 0$ implies $s = 0$. That is, no probability is an exact sum or difference of the others. We note that a vector drawn uniformly on $\Delta^{K-1}$ is incoherent with probability 1. For incoherent probability vectors, we can show universality of c-SGNNs. Moreover, this is actually a case where we *can* prove that c-GNNs are, in turn, *not* universal.

**Proposition 3.** *For one-hop input filtering, if $P$ is incoherent the space of functions $W \to \bar{\Psi}_{W,P}$ is dense in $\mathcal{C}(S_K, \mathbb{R})$. Moreover, there exists $P$ incoherent and $W \neq W'$ such that, for any c-GNN, $\bar{\Phi}_{W,P} = \bar{\Phi}_{W',P}$.*

**Additive kernel.** Let us now fix $W$ and examine universality with respect to $P$. A classical theorem on symmetric continuous functions [48] states that any $W$ can be arbitrarily well approximated as $W(x, y) \approx u(v(x) + v(y))$ for some functions $u, v$. Inspired by this result, a kernel will be said to be **additive** if it can (exactly) be written as $W(x, y) = u(v(x) + v(y))$, and **injectively additive** if both $u, v$ are *continuous and injective*. We prove universality in the unidimensional case below.

**Proposition 4.** *Let $\mathcal{X} \subset \mathbb{R}$, and $\tilde{\mathcal{P}}$ be any compact subset of $\mathcal{P}$. Assume $W$ is injectively additive with $\mathrm{Im}(v) \subset \mathbb{R}$. For one-hop filtering, the space of functions $P \to \bar{\Psi}_{W,P}$ is dense in $\mathcal{C}(\tilde{\mathcal{P}}, \mathbb{R})$.*

It is easy to see that injectively additive kernels include all SBMs for which $W_{ij} \neq W_{ij'}$ when $j \neq j'$, so Prop. 4 completes Prop. 3. However, unlike additive kernels, injectively additive kernels are *a priori* **not** universal approximators of symmetric continuous functions: this result [48] is only valid when $\mathrm{Im}(v)$ can be multidimensional. But it is known for instance that there is no continuous injective map from $[0, 1]^2$ to $[0, 1]$, so if $\mathrm{Im}(v) = [0, 1]^2$, $u$ cannot be both continuous and injective.

**Radial kernel.** We conclude this section with an important class of kernels, *radial* kernels $W(x, y) = w(\|x - y\|)$ for some function $w : \mathbb{R}_+ \to [0, 1]$. They include the popular Gaus-

sian kernel and so-called $\varepsilon$-graphs. Below, we give an example in one dimension, for which c-SGNNs are universal on symmetric distributions. The case of non-symmetric distributions seems more involved and we leave it for future investigations.

**Proposition 5.** *Assume that $\mathcal{X} = [-1, 1]$ and $W(x, y) = w(|x - y|)$ where $w$ is continuous and injective. Let $\tilde{\mathcal{P}} \subset \mathcal{P}$ be any compact set of **symmetric** distributions. For one-hop input filtering, the space of functions $P \to \bar{\Psi}_{W,P}$ is dense in $\mathcal{C}(\tilde{\mathcal{P}}, \mathbb{R})$.*

## 5 Approximation power: permutation-equivariant case

In the equivariant case, recall that the outputs of c-(S)GNNs are *functions* on $\mathcal{X}$. The "traditional" notion of universality is to evaluate the approximation power of mappings $(W, P) \to \mathcal{F}$, where $\mathcal{F}$ is some space of equivariant functions, as is done for the discrete case in [23, 3]. However, a potentially simpler and more relevant notion here is to *fix* $(W, P)$, and directly examine the properties of the space of functions $\mathcal{X} \to \mathbb{R}$ represented by c-(S)GNNs, that is, the space of functions $\{\Psi_{W,P} : \mathcal{X} \to \mathbb{R}\}$ for all possible c-SGNNs $\Psi_{W,P}$ (and similar for $\Phi_{W,P}$). Indeed, this directly answers such questions as: given an SBM, does there exist a c-GNN that can labels the communities (Fig. 1)? Or: given the structure of a mesh, what functions can be computed on it, e.g. for segmentation?

### 5.1 Generic result with Stone-Weierstrass theorem

As in the invariant case, the Stone-Weierstrass theorem yields a generic separation condition.

**Proposition 6.** *Let $(W, P)$ be fixed. Then c-SGNNs (resp. c-GNNs) are dense in $\mathcal{C}(\mathcal{X}, \mathbb{R})$ iff: for all $x \neq x' \in \mathcal{X}$, there is a c-SGNN (resp. a c-GNN) such that $\Psi_{W,P}(x) \neq \Psi_{W,P}(x')$ (resp. $\Phi_{W,P}(x) \neq \Phi_{W,P}(x')$).*

If $(W, P)$ are such that c-(S)GNNs satisfy some symmetry or invariance (see for instance Prop. 9), then the space $\mathcal{X}$ can be quotiented to obtain universality among functions satisfying these constraints.

### 5.2 c-SGNNs are more powerful than c-GNNs

Using a proof similar to Theorem 3, we can then prove that c-SGNNs are indeed strictly more powerful than c-GNNs for some $(W, P)$.

**Theorem 4.** *For both one- and two-hop input filtering, the following holds. For any $(W, P)$, the set of functions of the form $\Phi_{W,P}$ is included in the set of functions $\Psi_{W,P}$, and there exist $(W, P)$ such that the inclusion is **strict**.*

Again, we note that, for some random graph models $(W, P)$, c-SGNNs and c-GNNs might have the same approximation power.

### 5.3 Examples

We treat the same examples as before, with the addition of two-hop filtering for SBMs and radial kernels on the $d$-dimensional sphere.

**SBM.** Universality in the SBM case corresponds to being able to distinguish communities, that is, $\Psi_{W,P} : \mathcal{X} \to \mathbb{R}$ returns a different value for each element of the latent space $\mathcal{X}$. In the following result, we assume that $W$ is invertible, and prove the result for *both* one- and two-hop filtering, meaning that c-SGNNs can indeed distinguish communities of SBMs with random edges under some mild conditions. On the other hand, c-GNNs may fail on such models.

**Proposition 7.** *Let $P$ be incoherent, and $W$ be invertible. For **both** one- and two-hop input filtering, c-SGNNs are dense in $\mathcal{C}(\mathcal{X}, \mathbb{R})$. Moreover, there exist $(W, P)$ satisfying the conditions above such that any c-GNN $\Phi_{W,P}$ is a constant function.*

This proposition is illustrated in Fig. 1: on an SBM with constant degree function, any GNN converges to a c-GNN with constant output, while a SGNN with two-hop input filtering can be close to a c-SGNN that perfectly separates the communities.

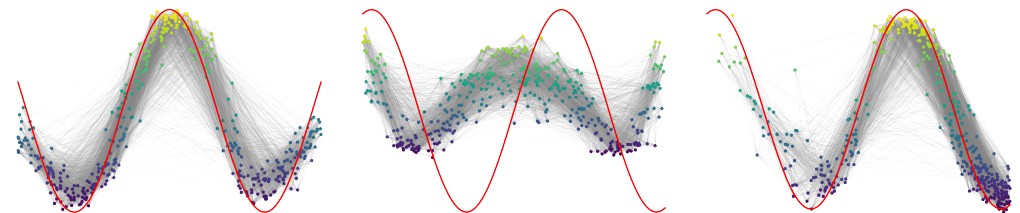

Figure 3: Illustration of Prop. 9 on Gaussian kernel, but with random edges and two-hop input filtering. The x-axis is the latent variables in $\mathcal{X} = [-1, 1]$. The y-axis is the output of a SGNN trained to approximate some function $f$ (red curve). On the left, both distribution $P$ and $f$ are symmetric, c-SGNNs are universal in that case. In the center, $P$ is symmetric but not $f$, and the training expectedly fails since the limit c-SGNN is symmetric. On the right, $P$ and $f$ are non-symmetric, and universality holds again. Details can be found in App. E.

**Additive kernel.** As in the invariant case, injectively additive kernels lead to universality.

**Proposition 8.** *Assume $W$ is injectively additive, fix any $P$. Then, for one-hop input filtering, c-SGNNs are dense in $\mathcal{C}(\mathcal{X}, \mathbb{R})$.*

**Radial kernel.** Unlike the invariant case (Prop. 5) which was limited to symmetric distributions, we treat both symmetric and non-symmetric case: when $P$ is symmetric, then so is $\Psi_{W,P}$ (by permutation-equivariance), and we have universality *among symmetric functions*. When $P$ is non-symmetric, we have universality among *all* functions. See Fig. 3 for an illustration.

**Proposition 9.** *Consider $\mathcal{X} = [-1, 1]$, $W(x, y) = w(|x - y|)$ a radial kernel with an invertible, continuous $w$, and $P$ with a piecewise continuous density such that $\mathbb{E}_P X = 0$. Then, for one-hop input filtering: if $P$ is symmetric, c-SGNNs are dense in the space of* symmetric *functions in $\mathcal{C}(\mathcal{X}, \mathbb{R})$, and if $P$ is not symmetric, c-SGNNs are dense in $\mathcal{C}(\mathcal{X}, \mathbb{R})$.*

Finally, we look at radial kernels on the $d$-dimensional sphere $\mathcal{X} = \mathbb{S}^{d-1} = \{x \in \mathbb{R}^d \mid \|x\| = 1\}$, an important example sometimes referred to as random geometric graphs [34, 2], or dot-product kernels [33], which are for instance popular in social networks analysis [47]. Indeed, in this case, the kernel only depends on the dot product $W(x, y) = w(x^\top y)$. Denoting by $d\tau$ the uniform measure on $\mathbb{S}^{d-1}$, it is known [14, 9] that functions in $L^2(d\tau)$ can be uniquely decomposed as $f(x) = \sum_{k \geqslant 0} f_k(x) = \sum_{k \geqslant 0} \sum_{j=1}^{N(d,k)} a_{k,j} Y_{k,j}(x)$ where $Y_{k,j}$ are *spherical harmonics*, that is, homogeneous harmonic polynomials of degree $k$ which form an orthonormal basis of $L^2(d\tau)$. We will say that such a function is **injectively decomposed** if the mapping $x \to [f_k(x)]_{k \geqslant 0}$ from $\mathbb{S}^{d-1}$ to $\ell_2(\mathbb{R})$ is injective. Note that generically this is verified if $f_k$ is non-zero for more than $d - 1$ distinct values of $k > 0$, as this corresponds to solving an over-determined system of polynomial equations, but there may be degenerate situations where this is not enough. The proof of the following proposition is based on the well-known Legendre/Gegenbauer polynomial decomposition of spherical harmonics [9] (see App. C.5).

**Proposition 10.** *Assume that $\mathcal{X} = \mathbb{S}^{d-1}$, that $W(x, y) = w(x^\top y)$ with continuous invertible $w : [-1, 1] \to [0, 1]$, and that $P = f d\tau$ has a density $f$ which is injectively decomposed. Then for one-hop input filtering c-SGNNs are dense in $\mathcal{C}(\mathcal{X}, \mathbb{R})$.*

## 6 Conclusion and outlooks

It is known that permutation-invariant GNNs fail to distinguish regular graphs of the same order, and permutation-equivariant GNNs return constant output on regular graphs. Similarly, their continuous counterparts suffer from the same flaw on random graph with constant or almost-constant degree function [30]. However, we showed that the recently proposed SGNNs converge to continuous architectures which, like in the discrete world, are strictly more powerful than c-GNNs. Moreover, we proved that both permutation-invariant and permutation-equivariant c-SGNNs are universal on many random graph models of interest, including a large class of SBMs and random geometric graphs.

We believe that our work opens many possibilities for future investigations. We examined very simple strategies for choosing the inputs $E_q(A)$ of the SGNN, but more complex, spectral-based choices

could exhibit better convergence properties and approximation power. We showed universality in specific random graph models of interest, but deriving a more generic criterion is still an open question. More directly, most of our examples illustrate the one-dimensional case $\mathcal{X} \subset \mathbb{R}$, and a generalization to multidimensional latent spaces would be an important step forward. Besides SGNNs, architectures that include high-order tensors [31] (sometimes called FGNN [3]) are known to be more powerful than the WL test. Conditions for their convergence on large graphs are still open, in particular since high-order tensors lead to high-order operators that may be difficult to manipulate. Finally, we remark that directly estimating the latent variables $x_i$ is a classical task in statistics, for which conditions of success have been derived for various approaches, e.g. for Spectral Clustering [25]. Comparing them with (S)GNNs is an important path for future work.

## Acknowledgement

This work was partly supported by ANR GraVa ANR-18-CE40-0005.

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
