# A Convergence

Let us start with some notations. Given a GNN $\Phi$, we define some bounds on its parameters that will be used in the multiplicative constants of the theorem. Recall that the filters are written $h_{ij}^{(\ell)}(\lambda) = \sum_{k=0}^{\infty} \beta_{ijk}^{(\ell)} \lambda^k$. We define $B_k^{(\ell)} = \left(\beta_{ijk}^{(\ell)}\right)_{ji} \in \mathbb{R}^{d_{\ell+1} \times d_\ell}$ the matrix containing the order-$k$ coefficients, and by $B_{k,|\cdot|}^{(\ell)} = \left(\left|\beta_{ijk}^{(\ell)}\right|\right)_{ji}$ the same matrix with absolute value on all coefficients. Recall that $\|\cdot\|$ is the operator norm for matrices. We define the following bounds:

$$H_2^{(\ell)} = \sum_k \left\|B_k^{(\ell)}\right\| \qquad\qquad H_{\partial,2}^{(\ell)} = \sum_k \left\|B_k^{(\ell)}\right\| k$$

$$H_\infty^{(\ell)} = \left\|B_{0,|\cdot|}^{(\ell)}\right\| + \sum_{k \geqslant 1} \left\|B_k^{(\ell)}\right\| \qquad\qquad H_{\partial,\infty}^{(\ell)} = \sum_k \left\|B_k^{(\ell)}\right\| k\sqrt{\log k}$$

which all converge by our assumptions on the $\beta_k$. We may also denote $H_2$ by $H_{L^2(P)}$ for convenience but this quantity does not depend on $P$. Note that, only for $H_\infty$, we use the spectral norm of the matrix $B_{0,|\cdot|}$ with non-negative coefficients, which is suboptimal compared to using $B_0$. This is due to a part of our analysis where we do not operate in a Hilbert space but only in a Banach space $\mathcal{B}(\mathcal{X})$, see Lemma 4. We also define $\left\|b^{(\ell)}\right\| = \sqrt{\sum_j (b_j^{(\ell)})^2}$ to measure the norm of the bias.

Given $X = \{x_1, \ldots, x_n\}$ and any dimension $d$, we denote by $S_X$ the sampling operator acting on functions $f : \mathcal{X} \to \mathbb{R}^d$ defined by $S_X f \overset{\text{def.}}{=} [f(x_1), \ldots, f(x_n)] \in \mathbb{R}^{n \times d}$. We have $\left\|\frac{1}{\sqrt{n}} S_X f\right\|_F \leqslant \|f\|_\infty$. Finally, given $X$ and $W$, we define $W(X) \overset{\text{def.}}{=} (W(x_i, x_j))_{ij} \in \mathbb{R}^{n \times n}$, and remark that $\frac{W(X)}{n} \circ S_X = S_X \circ T_{W,X}$. In the deterministic edges case, the adjacency matrix $A$ is directly $W(X)$. In the random edges case, $A$ has expectation $W(X)$ (conditionally on $X$).

## A.1 Convergence of GNNs: proof of Theorem 1

This proof is a variant of [22]. We prove Theorem 1 with the following error terms:

$$R_1(n) = \frac{C_1\sqrt{d_\mathcal{X}} + C_2\sqrt{\log\left(\frac{\sum_\ell d_\ell}{\rho}\right)}}{\sqrt{n}} \qquad R_2(n) = \frac{C_\nu C_3}{\sqrt{\alpha_n n}} \qquad R_3(n) = C_4\sqrt{\frac{\log(1/\rho)}{n}}, \quad (11)$$

where $R_2$ is present in the random edges case and $R_3$ in the permutation-invariant case, and the following constants:

$$C = L_g \prod_{\ell=0}^{M-1} H_2^{(\ell)} \qquad\qquad C_1 = \sum_{\ell=0}^{M-1} C^{(\ell)} H_{\partial,\infty}^{(\ell)}$$

$$C_2 = C_1(1 + D_\mathcal{X} L_W) \qquad\qquad C_3 = \sum_{\ell=0}^{M-1} C^{(\ell)} H_{\partial,2}^{(\ell)} \qquad\qquad C_4 = C^{(M)}$$

where the MLP $g$ is $L_g$-Lipschitz and $D_\mathcal{X} = \sup_{x,x' \in \mathcal{X}} m_\mathcal{X}(x, x')$ is the diameter of $\mathcal{X}$, and

$$C^{(\ell)} = L_g \left(\prod_{s=\ell+1}^{M-1} H_2^{(s)}\right) \left(C_f \prod_{s=0}^{\ell-1} H_\infty^{(s)} + \sum_{s=0}^{\ell-1} \left\|b^{(s)}\right\| \prod_{p=s+1}^{\ell-1} H_\infty^{(p)}\right)$$

with the conventions that an empty product is 1 and an empty sum is 0.

We begin the proof by the equivariant case, the invariant case will simply use an additional concentration inequality. We have

$$\text{MSE}_X\left(\Phi_A(Z^{(0)}), \Phi_{W,P}(f)\right) = \frac{1}{\sqrt{n}} \left\|\Phi_A(Z^{(0)}) - S_X \Phi_{W,P}(f^{(0)})\right\|_F$$

$$\leqslant L_g \frac{1}{\sqrt{n}} \left\|Z^{(M)} - S_X f^{(M)}\right\|_F.$$

We therefore seek to bound that last term. Define the notation

$$\Delta^{(\ell)} = \frac{1}{\sqrt{n}} \sqrt{ \sum_j \left\| \sum_i \left( h_{ij}^{(\ell)}\left(\frac{A}{n}\right) S_X f_i^{(\ell)} - S_X h_{ij}^{(\ell)}(T_{W,P}) f_i^{(\ell)} \right) \right\|^2 }.$$

Then, using the Lipschitzness of $\rho$, Lemma 4 with $\|A/n\| \leqslant 1$, and the fact that $S_X \circ \rho = \rho \circ S_X$, we have

$$\left\| Z^{(\ell+1)} - S_X f^{(\ell+1)} \right\|_F$$

$$= \left( \sum_j \left\| \rho\left( \sum_{i=1}^{d_\ell} h_{ij}^{(\ell)}\left(\frac{A}{n}\right) Z_{:,i}^{(\ell)} + b_j^{(\ell)} 1_n \right) - S_X \rho\left( \sum_{i=1}^{d_\ell} h_{ij}^{(\ell)}(T_{W,P}) f_i^{(\ell)} + b_j^{(\ell)} \right) \right\|^2 \right)^{\frac{1}{2}}$$

$$= \left( \sum_j \left\| \rho\left( \sum_{i=1}^{d_\ell} h_{ij}^{(\ell)}\left(\frac{A}{n}\right) z_i^{(\ell)} + b_j^{(\ell)} 1_n \right) - \rho\left( S_X \left( \sum_{i=1}^{d_\ell} h_{ij}^{(\ell)}(T_{W,P}) f_i^{(\ell)} + b_j^{(\ell)} \right) \right) \right\|^2 \right)^{\frac{1}{2}}$$

$$\leqslant \left( \sum_j \left\| \sum_{i=1}^{d_\ell} h_{ij}^{(\ell)}\left(\frac{A}{n}\right) Z_{:,i}^{(\ell)} - S_X h_{ij}^{(\ell)}(T_{W,P}) f_i^{(\ell)} \right\|^2 \right)^{\frac{1}{2}}$$

$$\leqslant \left( \sum_j \left\| \sum_{i=1}^{d_\ell} h_{ij}^{(\ell)}\left(\frac{A}{n}\right) \left( Z_{:,i}^{(\ell)} - S_X f_i^{(\ell)} \right) \right\|^2 \right)^{\frac{1}{2}}$$

$$+ \left( \sum_j \left\| \sum_{i=1}^{d_\ell} h_{ij}^{(\ell)}\left(\frac{A}{n}\right) S_X f_i^{(\ell)} - S_X h_{ij}^{(\ell)}(T_{W,P}) f_i^{(\ell)} \right\|^2 \right)^{\frac{1}{2}}$$

$$\leqslant H_2^{(\ell)} \left\| Z^{(\ell)} - S_X f^{(\ell)} \right\|_F + \sqrt{n} \Delta^{(\ell)}.$$

A recursion shows that, for all $Z^{(0)}$:

$$\mathrm{MSE}_X\left( \Phi_A(Z^{(0)}), \Phi_{W,P}(f) \right) \leqslant L_g \sum_{\ell=0}^{M-1} \Delta^{(\ell)} \prod_{s=\ell+1}^{M-1} H_2^{(s)}$$

$$+ L_g \left( \prod_{\ell=0}^{M-1} H_2^{(\ell)} \right) \mathrm{MSE}_X(Z^{(0)}, f^{(0)}). \tag{12}$$

We now bound all $\Delta^{(\ell)}$ with high probability. Recall that $\frac{W(X)}{n} \circ S_X = S_X \circ T_{W,X}$, and that we have $\left\| \frac{S_X}{\sqrt{n}} f \right\| \leqslant \|f\|_\infty$ and $\|T_{W,X}\|_\infty \leqslant 1$. By Lemma 4 we have

$$\Delta^{(\ell)} \leqslant \sqrt{ \sum_j \left\| \sum_i \left( h_{ij}^{(\ell)}\left(\frac{A}{n}\right) - h_{ij}^{(\ell)}\left(\frac{W(X)}{n}\right) \right) \frac{S_X}{\sqrt{n}} f_i^{(\ell)} \right\|^2 }$$

$$+ \sqrt{ \sum_j \left\| \sum_i \frac{S_X}{\sqrt{n}} \left( h_{ij}^{(\ell)}(T_{W,X}) - h_{ij}^{(\ell)}(T_{W,P}) \right) f_i^{(\ell)} \right\|^2 }$$

$$\leqslant H_{\partial,2}^{(\ell)} \left\| \frac{A - W(X)}{n} \right\| \left\| f^{(\ell)} \right\|_\infty$$

$$+ \sum_k \|B_k\| \sqrt{ \sum_i \left( \sum_{p=0}^{k-1} \left\| (T_{W,X} - T_{W,P}) T_{W,P}^{k-1-p} f_i^{(\ell)} \right\|_\infty \right)^2 }. \tag{13}$$

The first term in (13) is $0$ in the deterministic edges case. In the random edges case, it is handled with a recent concentration inequality for Bernoulli matrices [25], recalled in Theorem 5 in App. D. Since $\alpha_n \gtrsim \frac{\log n}{n}$, for any $\nu$, there is a constant $C_\nu$ such that, with probability $1 - n^{-\nu}$ on the random edges (conditionally on $X$), $\left\| \frac{A - W(X)}{n} \right\| \leqslant \frac{C_\nu}{\sqrt{\alpha_n n}}$. By the law of total probability, it is valid with joint probability $1 - n^{-\nu}$ on both $X$ and the random edges.

We now bound the second term in (13). Define $\rho_k = \frac{C\rho}{(k+1)^2 \sum_\ell d_\ell}$ with $C$ such that $\sum_{k\ell} d_\ell \rho_k = \rho/4$ (even when the filters are not of finite order). Using an application of Dudley's inequality detailed in Lemma 3, applied with $U(x, y) = W(x, y) f(y)$ which is bounded by $\|f\|_\infty$ and has Lipschitz constant $L_W \|f\|_\infty$ in the first variable, and a union bound, we obtain with probability $1 - \rho/4$ that: for all $i, \ell, k$, we have

$$
\left\| (T_{W,X} - T_{W,P}) T_{W,P}^k f_i^{(\ell)} \right\|_\infty \lesssim \frac{1}{\sqrt{n}} \left\| f_i^\ell \right\|_\infty \left( \sqrt{d_\mathcal{X}} + (1 + D_\mathcal{X} L_W) \sqrt{\log \rho_k^{-1}} \right).
$$

Coming back to the second term of (13), with probability $1 - \rho/4$:

$$
\sum_k \|B_k\| \sqrt{ \sum_i \left( \sum_{p=0}^{k-1} \left( \left\| (T_{W,X} - T_{W,P}) T_{W,P}^{k-1-p} f_i^{(\ell)} \right\|_\infty \right)^2 \right) }
$$

$$
\lesssim \frac{ \left( \sqrt{d_\mathcal{X}} + (1 + D_\mathcal{X} L_W) \sqrt{\log \frac{\sum_\ell d_\ell}{\rho}} \right) }{\sqrt{n}} \sum_k \|B_k\| k \sqrt{\log k} \sqrt{ \sum_i \left\| f_i^{(\ell)} \right\|_\infty^2 }
$$

$$
\leqslant \frac{ \left( \sqrt{d_\mathcal{X}} + (1 + D_\mathcal{X} L_W) \sqrt{\log \frac{\sum_\ell d_\ell}{\rho}} \right) }{\sqrt{n}} H_{\partial,\infty}^{(\ell)} \left\| f^{(\ell)} \right\|_\infty.
$$

At the end of the day we obtain that with probability $1 - \rho$, for all $\ell$:

$$
\Delta^{(\ell)} \propto \left\| f^{(\ell)} \right\|_\infty \left( \frac{H_{\partial,2}^{(\ell)}}{\sqrt{\alpha_n n}} + \frac{H_{\partial,\infty}^{(\ell)} \left( \sqrt{d} + (1 + D_\mathcal{X} L_W) \sqrt{\log \frac{\sum_\ell d_\ell}{\rho}} \right)}{\sqrt{n}} \right).
$$

We then use Lemma 5 to bound $\left\| f^{(\ell)} \right\|_\infty$ and conclude.

For the invariant case, we have

$$
\left\| \bar\Phi_A(Z^{(0)}) - \bar\Phi_{W,P}(f^{(0)}) \right\| \leqslant \mathrm{MSE}_X(\Phi_A(Z^{(0)}), \Phi_{W,P}(f^{(0)}))
$$

$$
+ L_g \left\| \frac{1}{n} \sum_{i=1}^n f^{(M)}(x_i) - \int f^{(M)}(x) dP(x) \right\|
$$

We use a vector Hoeffding's inequality [36, Lemma 4] and a bound on $\left\| f^{(M)} \right\|_\infty$ (Lemma 5) to conclude.

## A.2 Convergence of SGNNs

We prove Theorem 2 with the same form of error terms (11) where $R_i$ is replaced by $R_i'$ with modified multiplicative constants $C_i'$. Here we will have:

$$C' = DL_g \prod_{\ell=0}^{M-1} H_2^{(\ell)}$$

$$C_1' = DL_\Phi + \sum_{\ell=0}^{M-1} H_{\partial,\infty}'^{(\ell)} C'^{(\ell)} + \sum_{\ell=0}^{M-1} H_{\partial,\infty}^{(\ell)} C^{(\ell)}$$

$$C_2' = (1 + D_\mathcal{X} L_W) \sum_{\ell=0}^{M-1} H_{\partial,\infty}'^{(\ell)} C'^{(\ell)} + DL_\Phi D_\mathcal{X} + DC_\Phi + \sum_{\ell=0}^{M-1} H_{\partial,\infty}^{(\ell)} (C^{(\ell)} + D_\mathcal{X} L^{(\ell)})$$

$$C_3' = \sum_{\ell=0}^{M-1} H_{\partial,2}'^{(\ell)} C'^{(\ell)} + H_{\partial,2}^{(\ell)} C^{(\ell)}, \qquad C_4' = C'^{(M)}$$

where $H_\star'^{(\ell)}$ is like $H_\star^{(\ell)}$ but for the weights in $\Phi'$, the final-layer MLP of $\Phi'$ is denoted by $g'$ with a Lipschitz constant $L_{g'}$, and:

$$D = L_{g'} \prod_{\ell=0}^{M-1} H_2'^{(\ell)}$$

$$C'^{(\ell)} = L_{g'} \left( \prod_{s=\ell+1}^{M-1} H_2'^{(s)} \right) \left( C_\Phi \prod_{s=0}^{\ell-1} H_\infty'^{(s)} + \sum_{s=0}^{\ell-1} \left\| b'^{(s)} \right\| \prod_{p=s+1}^{\ell-1} H_\infty'^{(p)} \right)$$

$$C^{(\ell)} = DL_g \left( \prod_{s=\ell+1}^{M-1} H_2^{(s)} \right) \tilde{C}_\infty^{(\ell)}$$

$$L^{(\ell)} = DL_g \left( \prod_{s=\ell+1}^{M-1} H_2^{(s)} \right) \left( L_W \tilde{C}_\infty^{(\ell)} + \sqrt{d_\ell} L_\eta \prod_{s=0}^{\ell-1} H_\infty^{(s)} \right)$$

$$C_\Phi = \| g(0) \| + L_g \tilde{C}_\infty^{(M)}$$

$$L_\Phi = L_g \left( L_\eta \prod_{\ell=0}^{M-1} \left\| B_0^{(\ell)} \right\| + L_W \sum_{\ell=0}^{M-1} \left( \prod_{s=\ell+1}^{M-1} \left\| B_0^{(s)} \right\| \right) \tilde{C}_2^{(\ell)} \right)$$

with

$$\tilde{C}_\star^{(\ell)} = C_\eta \prod_{s=0}^{\ell-1} H_\star^{(s)} + \sum_{s=0}^{\ell-1} \left\| b^{(s)} \right\| \prod_{p=s+1}^{\ell-1} H_\star^{(p)} \quad \text{for } \star \in \{2, \infty\}.$$

We start by applying Theorem 1 on the outer GNN $\Phi'$. Since the result is uniformly valid over all input of the GNN $Z^{(0)}$ with probability $1 - \rho$:

$$\text{MSE}_X(\Psi_A, \Psi_{W,P}) \leqslant D\text{MSE}_X \left( \frac{1}{n} \sum_q \Phi_A(E_q(A)), \int \Phi_{W,P}(\eta(\cdot, x)) dP(x) \right) + R'(n) \quad (14)$$

where, from Theorem 1, $D = L_{g'} \prod_{\ell=0}^{M-1} H_2'^{(\ell)}$ is $C_1$ but for the weights in $\Phi'$, and $R'(n)$ is the error term formed by summing various $R_i'(n)$, taking into account that by Lemma 5 the function inputed in $\Phi'$ is bounded by $C_\Phi$.

We must therefore bound the first term in (14). We write

$$\mathrm{MSE}_X\left(\frac{1}{n}\sum_q \Phi_A(E_q(A)), \int \Phi_{W,P}(\eta(\cdot,x))dP(x)\right)$$

$$\leqslant \mathrm{MSE}_X\left(\frac{1}{n}\sum_q \Phi_A(E_q(A)), \frac{1}{n}\sum_q \Phi_{W,P}(\eta(\cdot,x_q))\right)$$

$$+ \mathrm{MSE}_X\left(\frac{1}{n}\sum_q \Phi_{W,P}(\eta(\cdot,x_q)), \int \Phi_{W,P}(\eta(\cdot,x))dP(x)\right)$$

$$\leqslant \sup_q \mathrm{MSE}_X\left(\Phi_A(E_q(A)), \Phi_{W,P}(\eta(\cdot,x_q))\right)$$

$$+ \left\|\frac{1}{n}\sum_q \Phi_{W,P}(\eta(\cdot,x_q)) - \int \Phi_{W,P}(\eta(\cdot,x))dP(x)\right\|_\infty. \tag{15}$$

Let us start with the second term. By Lemma 5 and the Lipschitzness of $g$, $U(x,y) \stackrel{\text{def.}}{=} \Phi_{W,P}(\eta(\cdot,y))(x)$ is $C_\Phi$-bounded and $L_\Phi$-Lipschitz with respect to $x$.

Hence, applying Lemma 3: with probability $1-\rho$,

$$\left\|\frac{1}{n}\sum_q \Phi_{W,P}(\eta(\cdot,x_q)) - \int \Phi_{W,P}(\eta(\cdot,x))dP(x)\right\|_\infty \lesssim \frac{L_\Phi\sqrt{d_\mathcal{X}} + (L_\Phi D_\mathcal{X} + C_\Phi)\sqrt{\log(1/\rho)}}{\sqrt{n}}. \tag{16}$$

For the first term in (15), we introduce some notations. We denote by $f_i^{(\ell)} : \mathcal{X} \times \mathcal{X} \to \mathbb{R}$ the bivariate function propagated at each layer of the inner part of the c-SGNN, as:

$$f_0^{(0)} = \eta \qquad\qquad f_j^{(\ell+1)} = \rho\left(\sum_i h_{ij}^{(\ell)}(T_{W,P})f_i^{(\ell)} + b_j^{(\ell)}\right) \tag{17}$$

where $T_{W,P}$ is here to be understood as an operator on $\mathcal{C}(\mathcal{X} \times \mathcal{X})$ defined by $T_{W,P}[f](x,y) = \int W(x,z)f(z,y)dP(z)$. With these notations, $\Phi_{W,P}(\eta(\cdot,x_q)) = g(f^{(M)}(\cdot,x_q))$. Note that we still have $\|T_{W,P}\|_\infty \leqslant 1$ for this version. We perform the computation as in the proof of Theorem 1 in (12) to obtain:

$$\sup_q \mathrm{MSE}_X\left(\Phi_A(E_q(A)), \Phi_{W,P}(\eta(\cdot,x_q))\right)$$

$$\leqslant L_g \sum_{\ell=0}^{M-1} \sup_q \Delta_q^{(\ell)} \prod_{s=\ell+1}^{M-1} H_2^{(s)} + L_g\left(\prod_{\ell=0}^{M-1} H_2^{(\ell)}\right)\sup_q \mathrm{MSE}_X(E_q(A), \eta(\cdot,x_q)) \tag{18}$$

with

$$\Delta_q^{(\ell)} = \frac{1}{\sqrt{n}}\sqrt{\sum_j \left\|\sum_i \left(h_{ij}^{(\ell)}\left(\frac{A}{n}\right)S_X f_i^{(\ell)}(\cdot,x_q) - S_X h_{ij}^{(\ell)}(T_{W,P})[f_i^{(\ell)}(\cdot,x_q)]\right)\right\|^2}. \tag{19}$$

Then, again we decompose

$$\sup_q \Delta_q^{(\ell)} \leqslant H_{\partial,2}^{(\ell)}\left\|\frac{A - W(X)}{n}\right\|\left\|f^{(\ell)}\right\|_\infty$$

$$+ \sum_k \|B_k\|\sqrt{\sum_i\left(\sum_{p=0}^{k-1}\left\|(T_{W,X} - T_{W,P})T_{W,P}^{k-1-p}[f_i^{(\ell)}]\right\|_\infty\right)^2} \tag{20}$$

where we recall here that $f_i^{(\ell)}$ is a bivariate function.

Again, the first term is $0$ in the deterministic edges case, and otherwise by Theorem 5 we have $\left\|\frac{A-W(X)}{n}\right\| \leqslant C_\nu/\sqrt{\alpha_n n}$ with probability $1 - n^{-\nu}$, and by Lemma 6 we have

$$\left\|f^{(\ell)}\right\|_\infty \leqslant C_\eta \prod_{s=0}^{\ell-1} H_\infty^{(s)} + \sum_{s=0}^{\ell-1} \left\|b^{(s)}\right\| \prod_{p=s+1}^{\ell-1} H_\infty^{(p)}.$$

Fix $k, \ell, i$ for now. We will apply Lemma 3 with $U : (\mathcal{X} \times \mathcal{X}) \times \mathcal{X} \to \mathbb{R}$ defined as $U((x, x'), y) = W(x, y)f(y, x')$ for $f(y, x') = T_{W,P}^k[f_i^{(\ell)}(\cdot, x')](y)$. Since $\|T_{W,P}\|_\infty \leqslant 1$ we have $\|f\|_\infty \leqslant \left\|f_i^{(\ell)}\right\|_\infty$. Then,

$$\begin{aligned}
\left\|T_{W,P}^k[f_i^{(\ell)}(\cdot, x')] - T_{W,P}^k[f_i^{(\ell)}(\cdot, x'')]\right\|_\infty &= \left\|T_{W,P}^k[f_i^{(\ell)}(\cdot, x') - f_i^{(\ell)}(\cdot, x'')]\right\|_\infty \\
&\leqslant \left\|f_i^{(\ell)}(\cdot, x') - f_i^{(\ell)}(\cdot, x'')\right\|_\infty \\
&\leqslant L_\eta m_\mathcal{X}(x, x') \prod_{s=0}^{\ell-1} H_\infty^{(s)}
\end{aligned}$$

by Lemma 6. Hence $U$ is bounded by $\left\|f_i^{(\ell)}\right\|_\infty$ and $L_i^{(\ell)}$-Lipschitz with respect to $(x, x')$, with

$$L_i^{(\ell)} = L_W \left\|f_i^{(\ell)}\right\|_\infty + L_\eta \prod_{s=0}^{\ell-1} H_\infty^{(s)}. \tag{21}$$

Finally, note that $\mathcal{X} \times \mathcal{X}$ is compact with covering numbers proportional to $\varepsilon^{-2d}$. Hence by Lemma 3 and a union bound, again defining $\rho_k$ as in the proof of Theorem 1 such that $\sum_{ik\ell} \rho_k = \rho$: with probability $1 - \rho$, we have simultaneously for all $i, k, \ell$:

$$\begin{aligned}
&\sup_x \left\|(T_{W,X} - T_{W,P})T_{W,P}^k f_i^{(\ell)}(\cdot, x)\right\|_\infty \\
&\qquad \lesssim \frac{1}{\sqrt{n}}\left(\left\|f_i^{(\ell)}\right\|_\infty \sqrt{d_\mathcal{X}} + (\left\|f_i^{(\ell)}\right\|_\infty + D_\mathcal{X} L_i^{(\ell)})\sqrt{\log \rho_k^{-1}}\right).
\end{aligned}$$

Hence, as in the previous proof:

$$\begin{aligned}
&\sum_k \|B_k\| \sqrt{\sum_i \left(\sum_{p=0}^{k-1}\left(\left\|(T_{W,X} - T_{W,P})T_{W,P}^{k-1-p} f_i^{(\ell)}\right\|_\infty\right)^2\right)} \\
&\qquad \leqslant \frac{\left(\left\|f^{(\ell)}\right\|_\infty \sqrt{d_\mathcal{X}} + (\left\|f^{(\ell)}\right\|_\infty + D_\mathcal{X} L^{(\ell)})\sqrt{\log \frac{\sum_\ell d_\ell}{\rho}}\right) H_{\partial,\infty}^{(\ell)}}{\sqrt{n}}
\end{aligned}$$

where $L^{(\ell)} = (\sum_i (L_i^{(\ell)})^2)^{\frac{1}{2}}$.

### A.3  Proof of Prop. 1

The error can be written as

$$\frac{1}{\sqrt{n}}\left\|A^2 e_q/n - S_X T_{W,P}(W(\cdot, x_q))\right\|_2 \leqslant \frac{1}{\sqrt{n}}\left\|A^2 e_q/n - W(X)^2 e_q/n\right\|_2 \\ + \|(T_{W,X} - T_{W,P})W(\cdot, x_q)\|_\infty.$$

Using chaining as in the previous section, we have

$$\sup_x \|(T_{W,X} - T_{W,P})W(\cdot, x)\|_\infty \lesssim \frac{1}{\sqrt{n}}\left(\sqrt{d_\mathcal{X}} + (1 + D_\mathcal{X} L_W)\sqrt{\log 1/\rho}\right).$$

The first term is 0 in the deterministic edges case, and otherwise:

$$\frac{1}{\sqrt{n}}\left\|A^2 e_q/n - W(X)^2 e_q/n\right\|_2 = \left(\frac{1}{n}\sum_i\left(\frac{1}{n}\sum_j a_{ij}a_{jq} - \frac{1}{n}\sum_j W(x_i,x_j)W(x_j,x_q)\right)^2\right)^{\frac{1}{2}}$$

$$\leqslant \left(\frac{1}{n}\sum_{i\neq q}\left(\frac{1}{n}\sum_j a_{ij}a_{jq} - W(x_i,x_j)W(x_j,x_q)\right)^2\right)^{\frac{1}{2}}$$

$$+ \frac{1}{\sqrt{n}}\left(\frac{1}{n}\sum_j a_{jq}^2 - \frac{1}{n}\sum_j W(x_j,x_q)^2\right)^2.$$

Now, by Bernstein inequality with

$$Var(a_{ij}a_{jq}) \leqslant \mathbb{E}(a_{ij}^2 a_{jq}^2) = \alpha_n^{-2}\mathbb{E}(a_{ij}a_{jq}) = \alpha_n^{-2}W(x_i,x_j)W(x_j,x_q)$$

for $i \neq q$ and a union bound, with proba $1-\delta$, we have:

$$\left|\frac{1}{n}\sum_j a_{ij}a_{jq} - \frac{1}{n}\sum_j W(x_i,x_j)W(x_j,x_q)\right| \lesssim \frac{\sqrt{\log(n/\delta)}}{\alpha_n\sqrt{n}} \text{ for all } q \text{ and } i \neq q.$$

Since $\left|\frac{1}{n}\sum_j a_{jq}^2 - \frac{1}{n}\sum_j W(x_j,x_q)^2\right| \leqslant 2$, we have

$$\frac{1}{\sqrt{n}}\sup_q\left\|A^2 e_q/n - S_X T_W(W(\cdot,x_q))\right\|_2 \lesssim \frac{\sqrt{\log(n/\delta)}}{\alpha_n\sqrt{n}} + \frac{\sqrt{d_{\mathcal{X}}} + (1 + D_{\mathcal{X}}L_W)\sqrt{\log 1/\rho}}{\sqrt{n}} \quad (22)$$

## B  Approximation power: invariant case

### B.1  Application of Stone-Weierstrass

*Proof of Prop. 2.* We do the proof for cSGNNws, it is exactly similar for cGNNs.

This is a direct application of Lemma 1: for any two cSGNNs $\bar{\Psi} : \mathcal{W} \times \mathcal{P} \to \mathbb{R}^d$, $\bar{\Psi}' : \mathcal{W} \times \mathcal{P} \to \mathbb{R}^{d'}$, their concatenation $[\bar{\Psi}, \bar{\Psi}'] : \mathcal{W} \times \mathcal{P} \to \mathbb{R}^{d+d'}$ is also a cSGNN (if they do not use the same input transforms $\eta, \eta'$, one can concatenate $\eta'' = [\eta, \eta']$), and for any MLP $g$, $g \circ \bar{\Psi}$ is also a cSGNN.

One must just check that cSGNNs are continuous with respect to $\|\cdot\|_\infty + \|\cdot\|_{\mathrm{TV}}$ on $\mathcal{W} \times \mathcal{P}$:

- $(W, P) \mapsto \eta$ is continuous by assumption ;
- for any $f_{W,P} \in \mathcal{C}(\mathcal{X}, \mathbb{R}^d)$ continuously indexed by $(W, P)$,

$$\|T_{W,P}[f_{W,P}] - T_{W',P'}[f_{W',P'}]\|_\infty \leqslant \|f_{W,P}\|_\infty (\|W - W'\|_\infty + \|P - P'\|_{\mathrm{TV}})$$
$$+ \|f_{W,P} - f_{W',P'}\|_\infty$$

  and similarly for $\left\|\int f_{W,P}dP - \int f_{W',P'}dP'\right\|$ ;
- the non-linearity $\rho$ is Lipschitz.

$\square$

### B.2  cSGNNs are more powerful than cGNNs

*Proof of Theorem 3.* By construction, cGNNs are included in cSGNNs, since one can take $\Phi = 0$ as the input GNN before pooling in (6).

To prove strict inclusion, we will construct two models $(W, P), (W', P')$ such that, for any cGNN we have $\bar{\Phi}_{W,P} = \bar{\Phi}_{W',P'}$, but there exists a cSGNN such that $\bar{\Psi}_{W,P} = \bar{\Psi}_{W',P'}$. We do the proof in the random edges case with two-hop input filtering $\eta_{W,P} = T_{W,P}(W)$, since such cSGNNs can of course also be constructed in the deterministic edges case.

Since $\mathcal{X}$ is not a singleton, one can can single out two arbitrary elements $x, x'$ and take $P$ as a sum of two Diracs over them, which is equivalent to considering that $\mathcal{X} = \{x, x'\}$ (since any invariant architecture involves a final integration by $P$, it is useless to consider $W$ outside of the support of $P$). This results in a two-community SBM, for which $P$ can be represented as a 2-vector on the simplex and $W$ as a 2-by-2 symmetric matrix. We then consider a family of SBMs indexed by $\gamma \in [0, 1]$:

$$P = \begin{pmatrix} 1/3 \\ 2/3 \end{pmatrix}, \quad W_\gamma = \begin{pmatrix} \gamma & \frac{1-\gamma}{2} \\ \frac{1-\gamma}{2} & \frac{1+\gamma}{4} \end{pmatrix}.$$

It is not hard to see that $T_{W_\gamma, P}[1] = 1/3 \cdot 1$ for any $\gamma$. Therefore, for any cGNN $\Phi$, the function propagated inside its layers is always constant, and does not depend on $\gamma$. That is, $\bar{\Phi}_{W_\gamma, P} = \bar{\Phi}_{W_0, P}$ for any $\gamma$. On the other hand, consider the following SGNN:

$$\bar{\Psi}_{W,P} = \int_x \int_y f(T_W(W(\cdot, y))) dP(y) dP(x)$$

where $f$ is an MLP. By the universality theorem, $f$ can approximate $x \to x^2$, for which we obtain:

$$\bar{\Psi}_{W_\gamma, P} \approx 1/16 * \gamma^4 - 1/12 * \gamma^3 + 1/24 * \gamma^2 - 1/108 * \gamma + 17/1296.$$

This is not a constant function, so we can always find $\gamma, \gamma'$ such that $\bar{\Psi}_{W_\gamma, P} \neq \bar{\Psi}_{W_{\gamma'}, P}$, which concludes the proof. $\qquad\square$

## B.3  SBMs

*Proof of Prop. 3.* We apply Prop. 2. We fix $P$ as an incoherent vector in the $k$-simplex, and define $\mathcal{M} = \{(W, P) : W \in S_k([0,1])\}$ which is indeed compact. It therefore suffices to show that cSGNNs separates points in $\mathcal{M}$.

We proceed by contraposition: assume that $W, W'$ are such that $\bar{\Psi}_{W,P} = \bar{\Psi}_{W',P}$ for any cSGNN $\Psi$. We must show that necessarily $W = W'$. We look at cSGNNs of the form

$$\Psi_{W,P} = \int f_1 \left( \int f_0 \left( W(x, y) \right) dP(y) \right) dP(x)$$

$$= \sum_i P_i f_1(\sum_j P_j f_0(W_{ij})) = \sum_i P_i f_1(\sum_j P_j f_0(W'_{ij}))$$

where $f_0, f_1$ are MLPs. By the universality theorem, they can approximate any continuous function. Pick any $f_0$. Then $f_1$ can be chosen as to take only values in $\{0, 1\}$ on the discrete set $\{\sum_j P_j f_0(W_{ij}), \sum_j P_j f_0(W'_{ij})\}_i$ of size $2K$. Moreover, if there was an index $i_0$ such that $\sum_j P_j f_0(W_{i_0j}) \neq \sum_j P_j f_0(W'_{i_0j})$, $f_1$ can be chosen to give different values on them. Then, defining $s_i = f_1(\sum_j P_j f_0(W_{ij})) - f_1(\sum_j P_j f_0(W'_{ij})) \in \{-1, 0, 1\}$, we have both $s_{i_0} \neq 0$ and

$$\sum_i P_i s_i = \sum_i P_i \left( f_1(\sum_j P_j f_0(W_{ij})) - f_1(\sum_j P_j f_0(W'_{ij})) \right) = 0$$

which contradicts the incoherence of $P$. So, for all $f_0$ and $i$, we have $\sum_j P_j f_0(W_{ij}) = \sum_j P_j f_0(W'_{ij})$. By the exact same reasoning on $f_0$, we obtain that for all $i, j$, $W_{ij} = W'_{ij}$, which concludes the proof.

For the failure of c-GNNs, the proof is immediate using the example SBM in the proof of Theorem 3, since $P = [1/3, 2/3]$ is indeed incoherent. $\qquad\square$

## B.4  Decomposed kernel

*Proof of Prop. 4.* Applying Prop. 2 with $\mathcal{M} = \{W\} \times \tilde{\mathcal{P}}$, it suffices to show that cSGNNs separate the distributions in $\tilde{\mathcal{P}}$. By contraposition, assume that $P, P' \in \tilde{\mathcal{P}}$ are such that $\bar{\Psi}_{W,P} = \bar{\Psi}_{W,P'}$ for any cSGNN, and we want to prove that necessarily $P = P'$.

We look at cSGNN of the form $P \mapsto \int f_1(\int f_0(W(x,y))dP(y))dP(x)$, where $f_0, f_1$ are MLPs, that can approximate any continuous functions by the universality theorem. Since $u$ is continuous and injective, it is well-known that it has a continuous inverse on its image. Hence $f_0$ can be chosen to approximate $f_0 \approx u^{-1}$. By choosing $f_1$ to approximate $x \to x^k$, we obtain that:

$$\int \left(v(x) + \mathbb{E}_P v\right)^k dP(x) = \int \left(v(x) + \mathbb{E}_{P'} v\right)^k dP'(x).$$

Taking $k = 1$ we obtain that $\mathbb{E}_P v = \mathbb{E}_{P'} v$, and by an easy recursion we have $\mathbb{E}_P v^k = \mathbb{E}_{P'} v^k$ for all $k$. Since $v$ is invertible and polynomial functions are universal approximators on compacts one can write $v^{-1}(x) = \sum_k a_k x^k$ and $x = \sum_k a_k v(x)^k$, such that $\mathbb{E}_P X^k = \mathbb{E}_{P'} X^k$. Again, by the universality of polynomial functions, $\mathbb{E}_P f = \mathbb{E}_{P'} f$ for any continuous function, which is well-known to be equivalent to $P = P'$ and concludes the proof. $\square$

### B.5  Radial kernel

*Proof of Prop. 5.* We proceed as in the proof of Prop. 4 above: assuming $P, P' \in \tilde{\mathcal{P}}$ are such that $\bar{\Psi}_{W,P} = \bar{\Psi}_{W,P'}$ for any cSGNN, we want to prove that necessarily $P = P'$. We look at cSGNN of the form $P \mapsto \int f_1(\int f_0(W(x,y))dP(y))dP(x)$, where $f_0, f_1$ are MLPs. Since $w$ is injective $f_0$ can approximate $(x \to x^2) \circ w^{-1}$. By choosing $f_1$ to approximate $x \to x^k$, and since $P, P'$ are centered we obtain

$$\int \left(x^2 + \mathbb{E}_P X^2\right)^k dP(x) = \int \left(x^2 + \mathbb{E}_{P'} X^2\right)^k dP'(x).$$

Taking $k = 1$ we have $\mathbb{E}_P X^2 = \mathbb{E}_{P'} X^2$, and by an easy recursion $\mathbb{E}_P X^{2k} = \mathbb{E}_{P'} X^{2k}$ for all $k$. Since $P, P'$ have 0 odd-order moments, $\mathbb{E}_P X^k = \mathbb{E}_{P'} X^k$ for all $k$, from which we can conclude $P = P'$ as in the previous proof. $\square$

## C  Approximation power: equivariant case

### C.1  Application of Stone-Weierstrass

*Proof of Prop. 6.* As the proof of Prop. 2, this is a direct application of Lemma 1: the set of cSGNNs is closed by concatenation and composition with an MLP, $\mathcal{X}$ is compact, and any equivariant cSGNN in continuous since by assumption $W$, $\eta_{W,P}$ and $\rho$ are. $\square$

### C.2  cSGNNs are more powerful than cGNNs

*Proof of Theorem. 4.* As in the proof of Theorem 3 in App. B.2, non-strict inclusion is immediate. To prove strict inclusion, as in the proof of Theorem 3 we consider again the same 2-community SBM but for $\gamma = 1/2$:

$$P = \begin{pmatrix} 1/3 \\ 2/3 \end{pmatrix}, \quad W = \begin{pmatrix} 1/2 & 1/4 \\ 1/4 & 3/8 \end{pmatrix}.$$

Again any c-GNN would return a constant function $\Phi_{W,P}(1) = \Phi_{W,P}(2)$, while if we consider the following c-SGNN for two-hop filtering:

$$\Psi_{W,P} = \int f(T_W(W(\cdot, y)))dP(y)$$

with $f$ an MLP that approximates $x \to x^2$, we obtain $\Psi_{W,P}(1) \approx 1/8$ and $\Psi_{W,P}(2) \approx 11/96$, hence a non-constant function. $\square$

### C.3  SBMs

*Proof of Prop. 7.* We treat the two-hop filtering case, since they are included in one-hop architectures. By Prop. 6, we must prove the separation of elements of $\mathcal{X}$, which here are discrete community labels $\mathcal{X} = \{1, \ldots, K\}$. Fix $P \in \Delta^{K-1}$ incoherent and $W \in S_K$ invertible. Assume that $k, k'$ are two

communities such that $\Psi_{W,P}(k) = \Psi_{W,P}(k')$ for all $\Psi$ with two-hop input filtering. We want to show that necessarily $k = k'$. By assumption, we have:

$$\sum_i f\left(\sum_j W_{kj}W_{ji}P_j\right)P_i = \sum_i f\left(\sum_j W_{k'j}W_{ji}P_j\right)P_i$$

for any MLP $f$. As in the proof of Prop. 3 in App. B.3, $f$ can approximate a function that is $\{0,1\}$-valued on its inputs such that, if there is an index $i$ such that $\sum_j W_{kj}W_{ji}P_j \neq \sum_j W_{kj}W_{ji}P_j$, then the incoherency of $P$ is contradicted. Hence, for all $i$, $\sum_j W_{kj}W_{ji}P_j = \sum_j W_{kj}W_{ji}P_j$, or in other words:

$$W \cdot (P \odot (W_{k,:} - W_{k',:})) = 0.$$

Since $W$ is invertible and $P$ has only non-zero coordinates (by incoherency), we obtain $W_{k,:} = W_{k',:}$. Since $W$ is invertible it has necessarily distinct columns, so $k = k'$, which concludes the proof.

For the failure of c-GNNs we use the example of the proof of Theorem. 4, for which $P$ is incoherent, $W$ is invertible, but any c-GNN is constant. $\quad\square$

### C.4 Additive kernel

*Proof of Prop. 8.* Again we apply Prop. 6. Assume that $x, x'$ are such that $\Psi_{W,P}(x) = \Psi_{W,P}(x')$ for all one-hop c-SGNN. In particular,

$$\int f\left(u(v(x) + v(y))\right)dP(y) = \int f\left(u(v(x') + v(y))\right)dP(y)$$

for all MLP $f$. By taking $f = u^{-1}$, we obtain $v(x) = v(x')$, which leads to $x = x'$ by assumption of injectivity and concludes the proof. $\quad\square$

### C.5 Radial kernel

*Proof of Prop. 9.* Again we apply Prop. 6. Let $x, x'$ such that $\Psi_{W,P}(x) = \Psi_{W,P}(x')$ for all c-SGNNs. We want to prove that: if $P$ is symmetric, then $x = x'$ or $x = -x'$ (i.e. we quotient $[-1,1]$ by symmetry), and if $P$ is not symmetric, then necessarily $x = x'$.

By assumption $\int f(W(x,y))dP(y) = \int f(W(x',y))dP(y)$ for all MLP $f$.

**If $P$ is symmetric.** By choosing $f = (\cdot)^2 \circ w^{-1}$, we have

$$0 = \mathbb{E}(x - X)^2 - \mathbb{E}(x' - X)^2 = x^2 + 2x\mathbb{E}X + \mathbb{E}X^2 - (x')^2 - 2x'\mathbb{E}X - \mathbb{E}X^2 = x^2 - (x')^2$$

which is indeed $x' = x$ or $x' = -x$.

**If $P$ is not symmetric.** By the previous reasoning, we still have $x' = x$ or $x' = -x$, however, we must now show that the case $x' = -x$ is not possible. By contradiction, assume $x' = -x$ (and $x \neq 0$). Denote $M_k = \mathbb{E}_P X^k$ the kth moment of $P$. By Lemma 2, $P$ is symmetric iff $M_{2k+1} = 0$ for all $k$. We are going to show that this is the case by recursion: that is true for $k = 0$ by assumption, and if $M_{2\ell+1} = 0$ for all $\ell \leq k - 1$, by taking $f_0(t) = t^{2k+2}$:

$$
\begin{aligned}
0 = \mathbb{E}(x - X)^{2(k+1)} - \mathbb{E}(x + X)^{2(k+1)} &= \sum_{\ell=0}^{2(k+1)} \binom{2(k+1)}{\ell} x^\ell (-1)^\ell M_{2(k+1)-\ell} \\
&\quad - \left(\sum_{\ell=0}^{2(k+1)} \binom{2(k+1)}{\ell} x^\ell M_{2(k+1)-\ell}\right) \\
&= \sum_{\ell=1}^{k+1} \binom{2(k+1)}{2\ell-1} x^{2\ell-1} M_{2(k+1-\ell)+1} \\
&= 2(k+1)x M_{2k+1}
\end{aligned}
$$

and therefore $M_{2k+1} = 0$, and $P$ is symmetric, which is a contradiction. Therefore, necessarily $x' = x$, which completes the proof. $\quad\square$

*Proof of Prop. 10.* Again we apply Prop. 6. Let $x, x' \in \mathbb{S}^{d-1}$ such that $\Psi_{W,P}(x) = \Psi_{W,P}(x')$ for all c-SGNNs. We want to prove that $x = x'$. In particular,

$$\int g(w(x^\top y))dP(y) = \int g(w(x'^\top y))dP(y), \tag{23}$$

for all MLP $g$.

Recall that we have assumed that that $P$ has a density $f$ decomposed as $f(x) = \sum_{k \geqslant 0} f_k(x) = \sum_{k \geqslant 0} \sum_{j=1}^{N(d,k)} a_{k,j} Y_{k,j}(x)$, where $Y_{k,j}$ are spherical harmonics.

Let $P_k$ denote the Legendre/Gegenbauer polynomial of degree $k$, which satisfies the addition formula

$$P_k(x^\top y) = \frac{1}{N(d,k)} \sum_{j=1}^{N(d,k)} Y_{k,j}(x) Y_{k,j}(y).$$

Then, taking $g = N(d,k) P_k \circ w^{-1}$, note that we have

$$\int g(w(x^\top y))dP(y) = N(d,k) \int P_k(x^\top y) f(y) d\tau(y)$$
$$= \sum_j Y_{k,j}(x) \langle f, Y_{k,j} \rangle_{L^2(d\tau)}$$
$$= f_k(x).$$

Thus, (23) implies $f_k(x) = f_k(x')$ for all $k$. By assumption of injectivity of $x \to [f_k(x)]_k$, necessarily $x = x'$, which concludes the proof. $\qquad\square$

## D  Additional material

**Lemma 1.** *Let $(\mathcal{X}, d)$ be a compact metric space, $\mathcal{F} \subset \cup_{d \geqslant 1} \mathcal{C}(\mathcal{X}, \mathbb{R}^d)$ be a subspace of continuous multivariate functions on $\mathcal{X}$ that is closed by concatenation, that is, $f, f' \in \mathcal{F} \Rightarrow [f, f'] \in \mathcal{F}$. Define $\mathcal{F}_\rho = \{g \circ f \mid f \in \mathcal{F}, \ g : \mathbb{R}^d \to \mathbb{R} \text{ is a MLP with non-linearity } \rho\}$, where $\rho$ is not polynomial. If $\mathcal{F}$ separates points, that is, $\forall x \neq x', \exists f \in \mathcal{F}, f(x) \neq f(x')$, then $\mathcal{F}_{\mathrm{MLP}}$ is dense in $\mathcal{C}(\mathcal{X}, \mathbb{R})$ for the supremum norm.*

*Proof.* The proof uses the classical Stone-Weierstrass theorem: an algebra of continuous functions that separates points is dense in the space of continuous functions (for the supremum norm).

The main point is to check that $\mathcal{F}_\rho$ is an algebra. It is closed by linear combination: for all $g, g'$ MLPs, there is a $g''$ such that $g \circ f + g' \circ f' = g'' \circ [f, f']$ and $\mathcal{F}$ is closed by concatenation. Closure by multiplication is not true in general, however, following [20], this is true when $\rho = \cos$: since $\cos(a)\cos(b) = \frac{1}{2}(\cos(a+b) - \cos(a-b))$, we have: for $g(x) = \sum_i a_i \cos(b_i^\top x + c_i)$ and similarly $g'$,

$$(g \circ f) \cdot (g' \circ f') = \sum_{ij} a_i a'_j \cos\left(b_i^\top f(x) + c_i\right) \cos\left((b'_j)^\top f'(x) + c'_j\right)$$
$$= \sum_{ij} a_i a'_j \frac{1}{2}\Big( \cos\left([b_i, b_j]^\top [f, f'](x) + c_i + c'_j\right)$$
$$- \cos\left([b_i, -b_j]^\top [f, f'](x) + c_i - c'_j\right) \Big)$$
$$= g'' \circ [f, f']$$

for a certain MLP $g''$.

Hence $\mathcal{F}_{\cos}$ is an algebra. Moreover, it separates points: for $x \neq x'$, by hypothesis there is a $f \in \mathcal{F}$ such that $f(x) \neq f(x')$, and by the universality theorem applied to MLPs, this is also true for some $g \circ f$.

To conclude the proof, we note that, by the universality theorem of MLPs, cos itself can be approached by a MLP with any non-polynomial non-linearity $\rho$, so that $\mathcal{F}_\rho$ is dense in $\mathcal{F}_{\cos}$. $\qquad\square$

**Lemma 2.** *A piecewise continuous function $p$ on $[-1, 1]$ is symmetric iff $\int t^{2k+1} p(t)dt = 0$ for all $k$.*

*Proof.* Recall that the Legendre polynomials $L_k$ of degree $k$ are such that: a) they form an orthogonal basis of piecewise continuous functions on $[-1, 1]$ for $L^2$, b) respect parity $L_k(-t) = (-1)^k L_k(t)$, c) involves only monomials of the same parity $t^{k-2p}$, $p = 0, \ldots, \lfloor \frac{k}{2} \rfloor$.

By considering the decomposition $p = \sum_k (\int L_k p) L_k$, it is immediate that $p$ is symmetric iff $\int L_{2k+1} p = 0$ for all $k$, which is the same as $\int t^{2k+1} p(t)dt = 0$ for all $k$. $\qquad\square$

**Lemma 3** (Chaining). *Let $(\mathcal{X}, m_{\mathcal{X}})$ be a compact metric space with diameter $D_{\mathcal{X}}$ and covering numbers $\mathcal{N}(\mathcal{X}, m_{\mathcal{X}}, \varepsilon) \propto \varepsilon^{-d_{\mathcal{X}}}$, and $\mathcal{Y}$ a measurable space. Consider a bivariate measurable function $U : \mathcal{X} \times \mathcal{Y} \to \mathbb{R}$ that is uniformly $C_U$-bounded, and $L_U$-Lipschitz in the first variable. Let $y_1, \ldots, y_n$ be drawn i.i.d from a distribution $P$ on $\mathcal{Y}$. Then, with probability at least $1 - \rho$,*

$$\left\| \frac{1}{n} \sum_i \eta(\cdot, y_i) - \int \eta(\cdot, y)dP(y) \right\|_\infty \lesssim \frac{L_U \sqrt{d_{\mathcal{X}}} + (L_U D_{\mathcal{X}} + C_U)\sqrt{\log(1/\rho)}}{\sqrt{n}}.$$

*Proof.* For any $x \in \mathcal{X}$, define

$$Y_x = \frac{1}{n} \sum_i U(x, x_i) - \int U(x, y)dP(y).$$

Since $|Y_x| \leqslant 2C_U$, for any fixed $x_0 \in \mathcal{X}$, by Hoeffding's inequality we have: with probability at least $1 - \rho$,

$$|Y_{x_0}| \lesssim C_U \sqrt{\frac{\log(1/\rho)}{n}}.$$

Now we have

$$\left\| \frac{1}{n} \sum_i U(\cdot, x_i) - \int U(\cdot, x)dP(x) \right\|_\infty = \sup_{x \in \mathcal{X}} |Y_x| \leqslant \sup_{x, x' \in \mathcal{X}} |Y_x - Y_{x'}| + |Y_{x_0}|.$$

The second term is bounded by the inequality above. For the first term, we are going to use Dudley's inequality "tail bound" version [40, Thm 8.1.6]. We first check the sub-gaussian increments of the process $Y_x$. The sub-gaussian norm $\|\cdot\|_{\psi_2}$ is defined in [40, Def. 2.5.6]. For any $x, x' \in \mathcal{X}$, we have

$$\begin{aligned}
\|Y_x - Y_{x'}\|_{\psi_2} &\lesssim \left\| \frac{1}{n} \sum_i U(x, y_i) - U(x', y_i) \right\|_{\psi_2} \\
&\lesssim \frac{1}{n} \left( \sum_{i=1}^n \|U(x, y_i) - U(x', y_i)\|_{\psi_2}^2 \right)^{\frac{1}{2}} \\
&\lesssim \frac{1}{n} \left( n \|U(x, \cdot) - U(x', \cdot)\|_\infty^2 \right)^{\frac{1}{2}} \\
&\leqslant \frac{L_U}{\sqrt{n}} m_{\mathcal{X}}(x, x')
\end{aligned}$$

where we have used, from [40], Lemma 2.6.8 for the first line, Prop. 2.6.1 for the second, Example 2.5.8 for the third, and the Lipschitz property of $U$ for the last.

Now, we apply Dudley's inequality [40, Thm 8.1.6] to obtain that with probability $1 - \rho$,

$$\begin{aligned}
\sup_{x, x' \in \mathcal{X}} |Y_x - Y_{x'}| &\lesssim \frac{L_U}{\sqrt{n}} \left( \int_0^1 \sqrt{\log \mathcal{N}(\mathcal{X}, d, \varepsilon)}d\varepsilon + D_{\mathcal{X}} \sqrt{\log(1/\rho)} \right) \\
&\lesssim L_U \frac{\sqrt{d} + D_{\mathcal{X}} \sqrt{\log(1/\rho)}}{\sqrt{n}}.
\end{aligned}$$

Combining with the decomposition above yields the desired result. $\qquad\square$

**Lemma 4** (Variant of Lemma 6 in [22]). *Let $(E, \|\cdot\|_E)$ be a Banach space and $(\mathcal{H}, \|\cdot\|_\mathcal{H})$ be a separable Hilbert space. Let $L, L'$ be two bounded operators on $E$, and $S : E \to \mathcal{H}$ be a linear operator such that $\|S\|_{\mathcal{H} \to E} \leqslant 1$. For $1 \leqslant i \leqslant d$ and $1 \leqslant j \leqslant d'$, let $h_{ij} = \sum_k \beta_{ijk} \lambda^k$ be a collection of analytic filters, with $B_k = (\beta_{ijk})_{ji} \in \mathbb{R}^{d' \times d}$ the matrix of order-$k$ coefficients, with operator norm $\|B_k\|$. Let $x_1, \ldots, x_d \in E$ be a collection of points. Then:*

$$\sqrt{\sum_j \left\| S \sum_i h_{ij}(L) x_i \right\|_\mathcal{H}^2} \leqslant \left( \sum_k \|B_k\| \|L^k\| \right) \sqrt{\sum_i \|x_i\|_E^2} \tag{24}$$

*and*

$$\sqrt{\sum_j \left\| S \sum_i (h_{ij}(L) - h_{ij}(L')) x_i \right\|_\mathcal{H}^2} \leqslant \sum_k \|B_k\| \sqrt{\sum_i \left( \sum_{\ell=0}^{k-1} \|L^\ell\| \|(L-L')(L')^{k-1-\ell} x_i\|_E \right)^2}. \tag{25}$$

*Now, if $\|Lx\|_E \leqslant \|Sx\|_\mathcal{H}$ for some Hilbert space $\mathcal{H}$, then*

$$\sqrt{\sum_j \left\| \sum_i h_{ij}(L) x_i \right\|_E^2} \leqslant \left( \|B_{0,|\cdot|}\| + \sum_{k \geqslant 1} \|B_k\| \|L^{k-1}\| \right) \sqrt{\sum_i \|x_i\|_E^2}. \tag{26}$$

*Proof.* The results (24) and (25) are directly from Lemma 6 in [22]. The result (26) is obtained by observing that

$$\sqrt{\sum_j \left\| \sum_i h_{ij}(L) x_i \right\|_E^2} = \sqrt{\sum_j \left\| \sum_{ik} \beta_{ijk} L^k x_i \right\|_E^2} \leqslant \sum_k \sqrt{\sum_j \left\| \sum_i \beta_{ijk} L^k x_i \right\|_E^2}$$

$$\leqslant \sqrt{\sum_j \left\| \sum_i \beta_{ij0} x_i \right\|_E^2} + \sum_{k \geqslant 1} \sqrt{\sum_j \left\| S \sum_i \beta_{ijk} L^{k-1} x_i \right\|_\mathcal{H}^2}.$$

We apply (24) on the second term and on the first:

$$\sqrt{\sum_j \left\| \sum_i \beta_{ij0} x_i \right\|_E^2} \leqslant \sqrt{\sum_j \left( \sum_i |\beta_{ij0}| \|x_i\|_E \right)^2} \leqslant \|B_{0,|\cdot|}\| \sqrt{\sum_i \|x_i\|_E^2}.$$

$\square$

**Lemma 5** (Properties of c-GNNs). *Apply a c-GNN to a random graph model. Denote by $f^{(\ell)}$ the function at each layer. Then we have*

$$\left\| f^{(\ell)} \right\|_* \leqslant \|f\|_* \prod_{s=0}^{\ell-1} H_*^{(s)} + \sum_{s=0}^{\ell-1} \left\| b^{(s)} \right\| \prod_{p=s+1}^{\ell-1} H_*^{(p)} \tag{27}$$

*where $*$ indicates $L^2(P)$ or $\infty$.*

*Moreover, for $x, x' \in \mathcal{X}$, we have*

$$\left\| f^{(\ell)}(x) - f^{(\ell)}(x') \right\| \leqslant \left( \prod_{s=0}^{\ell-1} \left\| B_0^{(s)} \right\| \right) \left\| f^{(0)}(x) - f^{(0)}(x') \right\|$$

$$+ L_W d_\mathcal{X}(x, x') \sum_{s=0}^{\ell-1} \left( \prod_{p=s+1}^{\ell-1} \left\| B_0^{(p)} \right\| \right) \left( \left\| f^{(0)} \right\|_{L^2(P)} \prod_{p=0}^s H_2^{(p)} + \sum_{p=0}^{s-1} \left\| b^{(p)} \right\| \prod_{r=p+1}^s H_2^{(r)} \right).$$

*Proof.* For $j \leqslant d_\ell$, using Lemma 4, the Lipschitzness of $\rho$ and the easy fact that $\|T_{W,P}f\|_\infty \leqslant \|f\|_{L^2(P)}$. we write

$$
\left\| f^{(\ell)} \right\|_* \leqslant \sqrt{ \sum_j \left\| \sum_{i=1}^{d_{\ell-1}} h_{ij}^{(\ell-1)}(T_{W,P}) f_i^{(\ell-1)} + b_j^{(\ell-1)} \right\|_*^2 }
$$

$$
\leqslant \sqrt{ \sum_j \left\| \sum_{i=1}^{d_{\ell-1}} h_{ij}^{(\ell-1)}(T_{W,P}) f_i^{(\ell-1)} \right\|_*^2 } + \left\| b^{(\ell-1)} \right\|
$$

$$
\leqslant H_*^{(\ell-1)} \left\| f^{(\ell-1)} \right\|_* + \left\| b^{(\ell-1)} \right\|.
$$

An easy recursion gives the result.

Now,

$$
\left\| f^{(\ell)}(x) - f^{(\ell)}(x') \right\| \leqslant \sqrt{ \sum_j \left( \sum_{i=1}^{d_{\ell-1}} h_{ij}^{(\ell-1)}(T_{W,P}) f_i^{(\ell-1)}(x) - h_{ij}^{(\ell-1)}(T_{W,P}) f_i^{(\ell-1)}(x') \right)^2 }
$$

$$
= \sum_k \left\| B_k^{(\ell-1)} \left[ T_{W,P}^k f_i^{(\ell-1)}(x) - T_{W,P}^k f_i^{(\ell-1)}(x') \right]_{i=1}^{d_{\ell-1}} \right\|
$$

and since $|T_{W,P}f(x) - T_{W,P}f(x')| \leqslant L_W d_{\mathcal{X}}(x, x') \|f\|_{L^2(P)}$ and $\|T_{W,P}\|_{L^2(P)} \leqslant 1$ by Schwartz inequality,

$$
\left\| f^{(\ell)}(x) - f^{(\ell)}(x') \right\| \leqslant \left\| B_0^{(\ell-1)} \right\| \left\| f^{(\ell-1)}(x) - f^{(\ell-1)}(x') \right\| + H_2^{(\ell-1)} \left\| f^{(\ell-1)} \right\|_{L^2(P)} L_W d_{\mathcal{X}}(x, x').
$$

Again we obtain the result by recursion. $\qquad\square$

**Lemma 6.** *(Properties of c-SGNNs) Denote by $f^{(\ell)}$ the bivariate functions propagated in the inner part of a c-SGNN. We have*

$$
\left\| f^{(\ell)} \right\|_\infty \leqslant C_\eta \prod_{s=0}^{\ell-1} H_\infty^{(s)} + \sum_{s=0}^{\ell-1} \left\| b^{(s)} \right\| \prod_{p=s+1}^{\ell-1} H_\infty^{(p)}.
$$

*Moreover,*

$$
\left\| f^{(\ell)}(\cdot, x) - f^{(\ell)}(\cdot, x') \right\|_\infty \leqslant L_\eta d_{\mathcal{X}}(x, x') \prod_{s=0}^{\ell-1} H_\infty^{(s)}.
$$

*Proof.* The first inequality is proved exactly as Lemma 5, noting that $\|T_{W,P}\|_\infty$ even for bivariate functions and $\|\eta\|_\infty \leqslant C_\eta$.

Then we have

$$
\left\| f^{(\ell)}(\cdot, x) - f^{(\ell)}(\cdot, x') \right\|_\infty \leqslant \sqrt{ \sum_j \left\| \sum_{i=1}^{d_{\ell-1}} h_{ij}^{(\ell-1)}(T_{W,P}) \left[ f_i^{(\ell-1)}(\cdot, x) - f_i^{(\ell-1)}(\cdot, x') \right] \right\|_\infty^2 }
$$

$$
\leqslant H_\infty^{(\ell-1)} \left\| f^{(\ell-1)}(\cdot, x) - f^{(\ell-1)}(\cdot, x') \right\|_\infty.
$$

$\qquad\square$

**Theorem 5** ([25])**.** *Let $A$ be a $n \times n$ symmetric matrix with independent Bernoulli entries $a_{ij} \sim \alpha_n p_{ij}$. Assume that $\alpha_n \gtrsim \frac{\log n}{n}$. Then, for all $\nu > 0$, there is a constant $C_\nu$ such that, for all $n$, with probability at least $1 - n^{-\nu}$:*

$$
\frac{1}{n} \left\| \frac{A}{\alpha_n} - P \right\| \leqslant \frac{C_\nu}{\sqrt{\alpha_n n}}. \tag{28}
$$

# E   Details of numerical experiments

**Figure 1.**   In this figure, we consider a 2-communities SBM with incoherent $P$, invertible $W$, but constant degree function. We use dense random edges with $\alpha_n = 1$. We train a permutation-equivariant GNN and a two-hop filtering SGNN on 5 random graphs with $n = 80$ nodes, output dimension $d_{out} = K$, with cross-entropy loss and the Adam optimizer. The displayed graph signal corresponds to the first dimension of the log-softmax of the ouput. The test graph has $n = 300$ nodes. The graph filters have order 1, such that we actually manipulate the message-passing version of GNNs. The GNN has $M = 5$ hidden layers with internal dimension $d_\ell = 250$ (except $d_0 = 1$ and $d_{out} = 2$) and is trained for 2000 epochs. Each of the GNNs constituting the SGNN has $M = 2$ hidden layers with dimension $d_\ell = 50$ and is trained for 1000 epochs.

**Figure 2.**   We compare one-hop and two-hop input filtering for a simple permutation-invariant SGNN, between the deterministic edges case and the random edges case. We know that the deterministic edges case converges to the c-SGNN in all settings, and we test if the random edges case converge to the deterministic one. We average over 50 random graphs with Gaussian kernel and a range of $n$'s with $\alpha_n \sim n^{-1/3}$, such that Prop. 1 applies in the two-hop case. The dominating term in the theoretical rate is $\mathcal{O}\left(1/(\alpha_n\sqrt{n})\right)$ from Prop. 1.

**Figure 3.**   Here we consider $\mathcal{X} = [-1, 1]$ with Gaussian kernel, and either a symmetric $P$ (uniform) or a non-symmetric but centered $P$ (here a well-adjusted affine by part function). We use random edges with $\alpha_n \sim n^{-1/3}$. We train a SGNN with two-hop input filtering to approximate either a symmetric function $x \to \cos(5x)$ or a non-symmetric one $x \to \sin(5x)$ with a simple square loss. We use 5 training graphs of size $n = 150$ and display a test graph with size $n = 400$.