# OpenReview forum: "On the Universality of Graph Neural Networks on Large Random Graphs"
_NeurIPS.cc/2021/Conference — NeurIPS 2021 Poster_

### Official Review · Reviewer_nJaH · 2021-07-01

**Rating:** 5
**Confidence:** 3

**Summary:**

This paper studies the universality of Structural Graph Neural Networks (SGNNs) and their continuous counterparts, termed c-SGNNs, in both the permutation-invariant and equivariant settings.  More specifically, the paper produces a theoretical convergence result between the recently proposed SGNNs and their continuous c-SGNNs in the limit of large graphs, analogously to existing connections between GNNs and their continuous counterparts c-GNNs. Then, the universality of SGNN is theoretically studied based on this convergence, and the expressive power of SGNNs is compared with that of standard GNNs. From this study, it is established that c-SGNNs are strictly more powerful than c-GNNs, and that c-SGNNs are universal on many random graph families, such as stochastic block models (SBMs), symmetric radial kernels, and injectively additive kernels. Finally, the overall universality of SGNNs and c-SGNNs, as well as open questions remaining in this regard, are discussed.

**Main Review:**

The paper proposes several theoretical contributions that can be of benefit to the GNN community. In particular, the definition of c-SGNN and the establishment of the correspondence between the two models produces novel insights about the power of SGNNs, and yields analogous theoretical knowledge to what is already established for standard GNNs. The study of model universality over distinct families of random graphs is also informative. However, I have concerns about the significance of the results and the overall contribution that these findings make. More specifically, I find that the results about universality over families of random graphs are of rather limited applicability and scope, as the results pertain to relatively niche classes of random graphs, which do not (to my knowledge) provide insights towards standard use cases of GNNs, unlike analogous WL and structure-based results for the discrete setting. Moreover, the results in this section appear to be disjoint, and there does not seem to be a means to leverage each result directly towards making more general discussions and deductions, or establishing a framework for studying SGNNs. Therefore, I feel that the paper would benefit from a better clarification of the scope of its results.

In terms of related work, the paper mentions higher-order GNNs and c-GNNs, but omits relevant works from the literature on the expressive power of GNNs that use randomness for allocating node features:

[1] R. Sato et al., "Random Features Strengthen Graph Neural Networks", SDM 2021.

[2] G. Dasoulas et al., "Coloring Graph Neural Networks for Node Disambiguation", IJCAI 2020.

[3] R. Abboud et al., "The Surprising Power of Graph Neural Networks with Random Node Initialization", IJCAI 2021.

These works achieve higher expressiveness (universality in [2] and [3]) and strong generalization simply by adding randomness to initial node features, and in fact randomly sampled features preserve invariance/equivariance in expectation and maintain the efficient running time of standard GNNs [3]. Therefore, these works should be presented in the paper and discussed against its contributions.

Finally, the writing of the paper could be improved with some changes, e.g., some basic concept and model definitions (like graphons) should be explicitly presented, graph neural networks can be presented more intuitively, the notation for invariant and equivariant functions should be made more distinctive (rather than a simple bar).

**Time Spent Reviewing:**

3.5

---

> ### Author Response · Authors · 2021-08-10
> **Response to Reviewer nJaH**
>
> Thank you for the detailed review.
> - **Clarification of the scope of our results**: we agree that the scope of the results deserves more explanation, and will elaborate on this in the paper. In this paper, our goal was primarily to characterize the power of GNNs of *large* graphs with many nodes. While graph isomorphism-based analyses like the WL test ones and their variants are appealing for relatively small graphs (eg molecules), large graphs are *never* exactly isomorphic between them, but present unique interesting features: communities of well-connected nodes, large-scale shapes from a sampled point cloud, etc. To model these macroscopic structures and analyze the behavior of GNNs on them, random graph models are classical tools with a very long history in the Statistics literature. Moreover, the considered (sparse) random graphs, sometimes called $W$-random graphs, latent space random graphs, among other names, include the vast majority of the random graphs literature (with the exception of preferential attachment models). They encompass most standard examples : Erdös-Rényi graphs, Stochastic Block Models, $\epsilon$-graphs (also called random geometric graphs), graphons, and so on. We will extend the introduction and add references in the final version to better clarify the scope of our result.
> - **Related work.** Thank you for providing these references, we will add them in the final version. To our understanding, these papers are in the same line of works as the papers by Loukas, stating that giving-up on permutation-invariance/equivariance allows for very powerful results, and we will extend the discussion on this topic. We also agree that the combination of the randomness on the GNN input introduced in these papers with the randomness of the random graph models could be a very fruitful avenue for future work.
> - **Writing.** Thank you for the suggestions, we will do our best to improve the readability of the paper, including definitions and intuitions related to graphons and GNNs.

---

### Official Review · Reviewer_uBKY · 2021-07-12

**Rating:** 8
**Confidence:** 3

**Summary:**

This paper studies continuous limits of Structured Graph Neural Networks (SGNNs), which authors call c-SGNNs. Prior work studied the continuous limits of GNNs as c-GNNs. Another prior work proposed SGNNs and showed it is more powerful than GNNs. This work merges these two results and show that SGNNs converge to c-SGNNs, and also that c-SGNNs are more powerful than c-GNNs. Specifically, c-GNNs do not cover Stochastic Block Models (SBMs), whereas c-SGNNs do.

**Limitations And Societal Impact:**

While injectively additive kernels and radial kernels are certainly important in the context they were proposed, it is unclear whether they can be interesting kernels for continuous limits of graphs. Maybe some discussions on the usefulness of these kernels on graph modeling context would strengthen the significance of the paper.

Although authors demonstrate c-SGNNs are more powerful than c-GNNs, somehow the counterexample seems quite different from that from SGNN paper. It would be interesting to see how these counterexamples are related. For example, would GNNs have trouble fitting graphs from SBMs?

**Main Review:**

Originality: Authors combine two recent innovations in GNNs. The first is c-GNN, the continuous limit of GNNs. The second is structured GNNs (SGNNs). Now authors study the continuous limit of SGNNs as c-SGNNs. The combination is not technically straightforward. SGNNs are based on one-hot encoding, but this does not have a straightforward continuous counterpart. Authors propose one-hop & two-hop filtering as alternatives, and study their convergence properties.

Quality & Significance: Approximation power analysis of Neural Networks is outside of my area of expertise, and I was not able to carefully verify proofs in the appendix. However, the narrative in the main paper certainly sounds technically plausible. The convergence analysis follows the strategy of Keriven et al (NeurIPS 2020). For power analysis, authors show that c-GNNs don't cover SBMs, while c-SGNNs do. This seems to be a significant observation, as SBMs are important class of random graph models. Also, authors also show that c-SGNNs are dense on injectively additive & radial kernels.

Clarity: The paper is certainly well-organized and easy to follow. Authors do a great job explaining high-level implications of each theorem/proposition. On the other hand, the main body of the paper discusses little about what technical challenges they had to deal with, and what proof techniques were used to solve them. The paper is quite rich with technical content, and therefore this is probably inevitable for a conference paper.

Significance: The mere fact that c-SGNNs are stronger than c-GNNs is not particularly surprising. But the fact that SBMs are counter examples of c-GNNs and c-SGNNs fix it is quite meaningful, considering the importance of SBMs in graph modeling literature. Also, authors introduce a few techniques that help analyze SGNNs: a spectral version of SGNN, and one/two hop filtering that replaces one-hot encoding.

**Time Spent Reviewing:**

6

---

> ### Author Response · Authors · 2021-08-10
> **Response to Reviewer uBKY**
>
> Thank you for your detailed review.
> - **Relevance of considered kernels.** We agree that “additive kernels” (and their “injective” version) sounds a bit technical, but we are not aware of a dedicated name in the literature. This definition is inspired by the universality theorem of the Deep Set paper by Zaheer et al., which states that all kernels (aka symmetric functions of two variables) can be arbitrarily well approximated by additive kernels $u(v(x)+v(y))$. While we currently need the additional injectivity condition for universality, we hope to find a way to treat the generic case in the future.
>
> - Radial kernels (sometimes called dot-product kernels) are increasingly popular in the context of random graphs, see for instance “Latent distance estimation for random geometric graphs” by Araya and de Castro at NeurIPS 2019 and reference therein, or their original definition in the PhD thesis of C Nickel “Random dot product graphs: A model for social networks”, for a detailed motivation behind their definitions. We will better motivate them in the final version and add a few references.
> - **Link with counterexample of Vignac et al.** The counter-examples between (c)GNNs and (c)SGNNs are actually related in the discrete and continuous cases. In the discrete case, Vignac et al. use a regular graph (that is, with constant degree) as an example of failure of the WL-test, on which SGNNs still succeed. Similarly, in the continuous case, we use an SBM with constant *degree function*. We will emphasize this in the final version.

---

### Official Review · Reviewer_5Rj4 · 2021-07-12

**Rating:** 7
**Confidence:** 3

**Summary:**

This paper considers structural Graph Neural Networks (SGNNs) which include unique node identifiers. The authors prove the convergence of SGNNs in the large graph limit to the continuous counterpart c-SGNNs. The c-SGNNs are proved to be more expressive than c-GNNs without the node identifiers. The universality is proved on some random graph models in both permutation-invariant and permutation-equivariant cases.

**Main Review:**

Originality: This paper presents a convergence analysis for SGNNs based on the previous convergence analysis for GNNs. It also presents a novel expressive power and universality analysis in both permutation-invariant and equivariant cases.

Quality/Significance: This paper provides sound theoretical analysis with specific cases considered separately. This is also a frontier work that focuses on the continuous limit of more powerful GNNs. The universality is analyzed from the kernel respective which is representative and meaningful. The limitation that I can see is that the authors do not illustrate the advantages of the c-SGNNs straightforwardly with numerical simulations. I feel it is better to stress more about the motivations and the potential application values. I also have a question in Section 3. In one-hop filtering it seems to be even if the graph is dense, i.e. $\alpha\sim 1$, MSE still diverges with a constant value. Therefore I am not clear why one-hop filtering could be used as a general strategy in section 4.

Clarity: The paper is overall well-organized and clear. I feel a little lost in section 4 among the parallel examples and propositions. I recommend illustrating one or two cases in particular with numerical experiments while including the other cases in the appendix. The sparsity factor $\alpha$ and $\alpha_n$ are inconsistent in the paper, which may be a typo.

**Time Spent Reviewing:**

4

---

> ### Author Response · Authors · 2021-08-10
> **Response to Reply 5Rj4**
>
> Thank you for your detailed review.
>
> - **Numerical simulations**: The numerical exploration of (c)(S)GNN was not the main priority of our work, and the figures included are mainly here to illustrate our theoretical findings. Moreover, the studied SGNNs have been proposed by Vignac et al. and benchmarked on real data in their paper. We will emphasize this in the final version.
>
> - **“I feel it is better to stress more about the motivations and the potential application values.”** Thank you for pointing this out. Our motivation is mainly to better characterize the power of GNNs and related architectures on *large* graphs, for which the existing WL-test-based analyses may not be well-adapted. We found out in this work that SGNNs are more powerful than GNNs from this point of view, and we hope our analysis might help to design even more powerful architectures in the future. We will elaborate further on this in the paper.
> One-hop filtering. You are correct to notice that, for one-hop filtering, the MSE only converges for deterministic edges, and not random edges, even for dense graphs. This was the reason why we introduced two-hop filtering, which converges for random edges and certain sparsity levels. Examples are sometimes valid for both cases and sometimes not, we will further clarify this in the paper. Finding other collections of input signals for SGNNs that lead to universality in the sparse random edges case is a major avenue for future work.
>
> - **"I feel a little lost in section 4 among the parallel examples and propositions"**: We will do our best to make this section easier to follow. In particular, we will give more insights on how Proposition 2 is the key to the following results, as a generalized Stone-Weierstrass consequence.
>
> - **"The sparsity factor $\alpha$ and $\alpha_n$ are inconsistent in the paper, which may be a typo."** This is indeed a typo, we will homogenize the notations in the final submission.

---

### Official Review · Reviewer_Q8cJ · 2021-07-14

**Rating:** 9
**Confidence:** 4

**Summary:**

The authors consider the SGNNs introduced in [36], they improve on the selection of unique node identifiers and extend the architecture to the limit of large random graphs. More importantly, the prove that by including the unique node identifiers, the resulting (c-)SGNN is more discriminative than the (c-)GNNs.

**Main Review:**

The paper is very interesting as it poses relevant theoretical results in the field of graph classification using GNNs. It is well written and it is technically solid. I, therefore, recommend it for acceptance at the conference.

My only comment, if the authors wish to consider it for future research in the area, relates to the nature of the permutation equivariance property of FIR graph filters. The authors state that GNNs with FIR graph filters are incapable of learning communities in an SBM model with the same degree. This is expected due to the permutation equivariance property of FIR graph filters. That is, FIR graph filters are designed to not being able to solve these problems, as this gives them the ability to learn faster and generalize better (as is the case with regular CNNs). Of course, the drawback is that FIR graph filters are not good at tasks in which permutation equivariance is a hassle (like is the case of identifying communities in an SBM). But they also should not be used for those cases. For cases in which permutation equivariance is a problem, there are many non-convolutional graph filters, like node-variant, or edge-variant (EdgeNets), that could work better in these problems. Maybe, instead of looking at unique node identifiers and how to adapt FIR graph filters to solve tasks that they are not suited for, one can just adopt a different kind of filter. Just an idea, that may be computationally less intensive (after all, unique node identifiers require additional N filters to be computed).

**Time Spent Reviewing:**

2

---

> ### Author Response · Authors · 2021-08-10
> **Response to Reviewer Q8cJ**
>
> We are glad that you enjoyed the paper! Thank you for your comment on FIR filters -- this seems like an interesting direction for us to look into in the future. Regarding the remark on the behavior of GNNs on SBMs, we will clarify that it is indeed due to permutation equivariance of the filters.

---

### Decision · Program_Chairs · 2021-09-27

**Decision:**

Accept (Poster)

**Comment:**

The paper studies structural GNNs in the large graph limit where they converge towards their continuous counterpart c-SGNN. The paper presents a variety of theoretical results showing that c-SGNNs are provably superior to the continuous version of vanilla GNNs namely, c-GNNs. In particular the paper shows that c-SGNNs can recover communities in a stochastic block models in regimes where c-GNNs fail.

All the reviewers agreed that the paper presents a novel and interesting set of theoretical results. However, the reviewers also felt that the paper can do a better job of discussing and comparing with existing literature. I recommend the paper for acceptance with a strong recommendation to the authors that they take into account the reviewer comments about existing work when preparing the camera ready version.